

# Single Footprint Retrievals for AIRS using a Fast TwoSlab Cloud-Representation Model and All-Sky Radiative Transfer Algorithm

Sergio DeSouza-Machado[1], L. Larrabee Strow[1,2], Andrew Tangborn[1], Xianglei Huang[3], Xiuhong Chen[3], Xu Liu[4], Wan Wu[5], and Qiguang Yang[5]

[1]JCET, University of Maryland Baltimore County, Baltimore, Maryland
[2]Dept of Physics, University of Maryland Baltimore County, Baltimore, Maryland
[3]University of Michigan, Ann Arbor, Michigan
[4]NASA Langley Research Center, Langley, Virginia
[5]Science and Systems and Applications, Inc, Hampton, Virginia

*Correspondence to:* Sergio DeSouza-Machado (sergio@umbc.edu)

**Abstract.** 1D-variational retrievals of temperature and moisture fields from hyperspectral infrared satellite sounders use cloud-cleared radiances as their observation. These derived observations allow the use of clear-sky only radiative transfer in the inversion for geophysical variables but at reduced spatial resolution compared to the native sounder observations. Cloud-clearing can introduce various errors, although scenes with large errors can be identified and ignored. Information content studies show

that when using multi-layer cloud liquid and ice profiles in infrared hyperspectral radiative transfer codes, there are typically only 2-4 degrees of freedom of cloud signal. This implies a simplified cloud representation is sufficient for some applications which need accurate radiative transfer. Here we describe a single-footprint retrieval approach for clear and cloudy conditions, which uses the thermodynamic and cloud fields from Numerical Weather Prediction (NWP) models as a first guess, together with a simple cloud representation model coupled to a fast scattering radiative transfer algorithm (RTA). The NWP model

thermodynamic and cloud profiles are first co-located to the observations, after which the N-level cloud profiles are converted to two slab clouds (typically one for ice and one for water clouds). From these, one run of our fast cloud representation model allows an improvement of the *a-priori* cloud state by comparing the observed and model simulated radiances in the thermal window channels. The retrieval yield is over 90%, while the degrees of freedom correlate with the observed window channel brightness temperature which itself depends on the cloud optical depth. The cloud representation/scattering package is bench-

marked against radiances computed using a Maximum Random Overlap cloud scheme. All-sky infrared radiances measured by NASA's Atmospheric Infrared Sounder (AIRS) and NWP thermodynamic and cloud profiles from the European Center for Medium Range Weather Forecasting (ECMWF) forecast model are used in this paper.

## 1   Introduction

Since the early 2000's, a number of high spectral resolution, low noise, very stable new generation hyperspectral infrared (IR)

sounders have been deployed on board Earth orbiting satellites, providing daily global Top of the Atmosphere (TOA) radiance





spectra. In principle these TOA radiances can be inverted to estimate atmospheric temperature and humidity profiles, minor gas concentration, surface temperature, and some clouds parameters.

IR sounders have rather large nadir footprints of ~15 km diameter, consequently far less than 10% of scenes are cloud free. Retrievals using eigenvalue regression methods have been used with these all-sky (cloud and clear) radiances (see for example Weisz et al. (2007)) but these methods have no reliable error estimates for individual scenes. Existing NASA and NOAA operational retrieval systems for IR sounders use cloud-clearing (Susskind et al., 1998, 2003) and (Gambacorta, 2013). In addition, Numerical Weather Prediction (NWP) centers generally only assimilate IR sounder radiances that have been deemed clear.

Presently the NASA-AIRS operational soundings are performed using cloud-cleared radiances coupled with a clear-sky RTA (Susskind et al., 1998, 2003). Cloud cleared radiances (CCRs) are synthesized using the differences in cloud amounts in a (typically) 3-by-3 set of adjacent Fields of View (FOVs) to produce a single effective estimate of the clear-sky radiance. This process increases the retrieval yield (to well above 10%) and provides some error estimates but simultaneously reduces the spatial resolution by a factor of three. Publicly available products from 1D variational retrievals include:

1. Atmospheric Infrared Sounder (AIRS): NASA, using cloud-clearing from 3x3 set of footprints (Susskind et al., 1998, 2003)

2. Cross Track Infrared Sounder (CrIS): The NOAA Unique Combined Atmospheric Processing System (NuCaps), also using cloud-clearing from 3x3 set of footprints (Gambacorta, 2013)

3. Infrared Atmospheric Sounding Interferometer (IASI): NOAA NuCaps, using cloud-clearing from 2x2 set of footprints (Gambacorta, 2013);

4. Infrared Atmospheric Sounding Interferometer (IASI): EUMETSAT, 2-step single-footprint retrievals: piecewise regression for all scenes nominally exploiting IASI in synergy with AMSU+MHS (IASI-only is fallback) followed by a physical retrieval using the Optimal Estimation Method (OEM) (Rodgers, 2000; Steck, 2001) on clear-scenes only (IASI-only) (August et al., 2012; EUMETSAT, 2016)

The retrieval approaches mentioned above use various combinations of training to NWP forecasts from the European Center for Medium Range Weather Forecasting (ECMWF) either by regression (IASI EUMETSAT) or with neural nets (AIRS NASA) or use climatology (CrIS NOAA). All utilize co-located microwave soundings when possible. The development of a formal error estimate computation in the NuCaps algorithm is underway (personal communication, Gambacorta). The CCR approaches lead to complicated quality control issues, since cloud-clearing can fail, and the decisions made in assigning quality flags to the retrievals are not trivial. The cloud-clearing process is especially problematic (Zhou et al., 2005) when the cloud fields are homogeneous and cloud-clearing becomes unstable and inaccurate, which introduce errors into retrieved products. This is not necessarily a problem for weather-forecasting oriented applications, since the retrieval Quality Assurance (QA) can accurately determine when cloud-clearing failed. The extensive QA in the AIRS retrieval system deems as many as ~20% of observations



as unsuitable for retrievals. This limits geographic sampling in a complicated way that could make these products problematic for climate statistics.

Single footprint retrievals with hyperspectral sounders could provide higher spatial resolution than the 3x3 cloud-clearing approach, which may be especially significant for water vapor due to its high spatial variability. They also are attractive since you are basing the retrieval on the observed quantity, the Level L1b (geolocated and calibrated) radiances, rather than a derived quantity, the cloud-cleared radiances. This requires a fast, and reasonably accurate scattering radiative transfer algorithm, where the cloud representation should be simple yet realistic enough to provide useful thermodynamic soundings. Single-footprint retrievals minimize the L1b observed minus computed brightness temperatures, unlike the AIRS Level 2 retrievals (atmospheric products derived from L1b radiances) which instead minimizes the difference between cloud-cleared radiances and computed brightness temperatures.

Here we examine some viable first steps in performing operational single-footprint retrievals using the OEM for these sensors using a fast scattering RTA that uses a first-guess (and *a-priori* estimates) from the ECMWF forecast model. The OEM formalism provides retrieval quality indicators, which give the user objective information in a natural way such as providing Averaging Kernels (AKs) and the number of Degrees of Freedom (DOF) in each retrieval, where it is especially important that single footprint retrievals return the *a-priori* below thick clouds.

Radiative Transfer Algorithms (RTAs) for the infrared that include scattering by clouds and aerosols are now available; see for example (Matricardi, 2005; Liu et al., 2009; De Souza-Machado et al., 2010; Liuzzi et al., 2016). These RTAs use accurate scattering algorithms, but initializing the cloud representation for retrievals is difficult and has not been used operationally.

This paper concentrates on the accuracy of our relatively simple, but fast accurate scattering model, especially when coupled with the representation of cloud features in the profile and initialization of these features in a retrieval. Very few cloud parameters can be retrieved from IR sounder spectra compared to clear-sky geophysical parameters (temperature and humidity), suggesting that simple fast scattering models and cloud representations should be sufficient to radiatively account for cloud/aerosol effects in a retrieval. The paper also demonstrates the utility of using NWP first guess model fields both for thermodynamic *and* cloud initialization in a *high yield* single footprint physical retrieval, where the computed degrees of freedom are shown to depend on the observed window channel brightness temperature (which itself depends on cloud loading).

In this paper observational data from AIRS is used, while the principal scattering algorithm is the Stand Alone Radiative Transfer Algorithm (SARTA) (Strow et al., 2003) for AIRS. AIRS was designed to provide improved temperature and humidity profiles for NWP and long-term climate studies. The AIRS radiances contain information about the thermodynamic state of the atmosphere (temperature, humidity), trace gases (such as ozone), surface parameters (Aumann and Pagano, 2002; Strow et al., 2003), as well as ice and water clouds (Kahn et al., 2003, 2005; Wu et al., 2009) and large aerosol particles (mineral dust and volcanic ash) (De Souza-Machado et al., 2010; Clarisse et al., 2010), though we do not consider aerosols in this paper. We introduce our cloud representation/scattering approach below, and test it statistically against an existing RTA, the Principal Component Radiative Transfer Model (PCRTM) (Liu et al., 2006, 2009) that has been supplemented with a full accounting of the cloud sub-grid variability (Maximum Random Overlap, or MRO) (Chen et al., 2013).



The PCRTM MRO implementation (Chen et al., 2013) uses the full vertical cloud profiles in the ECMWF model data. When a 50 sub-column MRO is added to the RTA to represent the cloud sub-grid variability, the radiance computation slows by 10X as compared to a 5 sub-column MRO. The appendix show that hyperspectral infrared radiances typically contain ~2-4 degrees of freedom of cloud information, which could be parametrized by the cloud amount, fraction, and cloud top and bottom

pressures. Our approach exploits this to reduce the cloud representation complexity from $N$-level model cloud fields for cloud ice water content and cloud liquid water content (and cloud cover) into two randomly overlapping slabs within the radiative transfer layers, greatly reducing the computational burden. The speed of the scattering calculations are then comparable to those under clear-sky conditions, and we show below that the radiances are as accurate as those from the MRO scheme.

The SARTA Two-Slab approach is then applied to single-footprint retrievals for an AIRS granule and compared to the

existing NASA AIRS Level 2 retrievals. As noted above, a key issue is the proper initialization of the cloud parameters in our RTA. Model fields from ECMWF are used here to initialize the thermodynamic and scattering cloud fields. Although NWP models do a reasonably good job at estimating cloud parameters, it is very unlikely that the positions of the model clouds are correct at scales near the sounder spatial resolution, especially given the time mis-match between available forecast models and the observations (±1.5 hours). Hence, our cloud parameters are chosen using the closest matches between simulated and

observed window region radiances, restricting the choices to model grid points close to the observation. This approach is key to the success of these single-footprint retrievals.

There are recent papers detailing hyperspectral Optimal Estimation based retrievals in the presence of clouds, see for example (Wu et al., 2017; Irion et al., 2017). Our approach is slightly different as it uses easily available NWP fields for initialization, and a simple cloud representation which allows for well defined jacobians to retrieve thermodynamic profiles and two cloud

decks, leading to high yields.

The paper is organized as follows. The AIRS instrument and the use of the ECMWF model are summarized first, followed by a detailed description of the RTA models and the cloud representation schemes. We then examine the computed radiance differences for both clear-sky and all-sky for these two RTAs and discuss radiance differences arising from perturbations to the TwoSlab cloud representation schemes. Finally we outline a method to reduce the impact of the spatial/ temporal mismatch

of observed versus modeled clouds, and use this together with the TwoSlab cloud representation to perform single footprint (cloudy) scene retrievals with an *a-priori* from the NWP model fields.

## 2  Background

### 2.1  The AIRS instrument and data

The Atmospheric Infrared Sounder (AIRS) on board NASA's polar orbiting EOS/Aqua platform has 2378 channels, covering

the Thermal Infrared (TIR) spectral range (roughly 649-1613 cm⁻¹ ) and shortwave infrared (2181-2665 cm⁻¹ ). The full widths at half maximum satisfy $\nu/\delta\nu \sim 1200$. The (spectral dependent) noise is typically $\leq 0.2$K. The instrument, operational since September 2002 is expected to continue operating until the early 2020's. In the comparisons presented later, we use about 1500 stable channels, avoiding channels that have deteriorated over time. AIRS has a 13.5 km nadir footprint from a ~705 km orbit,



and scans about ±49.5 degrees from nadir. Radiances from AIRS have been shown to be very stable and accurate (Aumann et al., 2006).

## 2.2 The ECMWF model fields

The core ECMWF 0-10 day forecasts are produced using the Integrated Forecasting System (IFS) (Uppala et al., 2005; Dee et al., 2011). The topmost ECMWF level is 0.01 mb, with terrain following $\sigma$-levels from the surface to 0.01 mb. The level spacing is finest in the boundary layer and coarsest at the top. At each model grid point, the cloud fields include the Cloud Water/Ice Water Content profiles (CIWC(z),CLWC(z) in g/g), Cloud Cover profile (CC(z)) and Total Cloud Cover (tcc).

Here we use 91 level ECMWF model fields, at a horizontal resolution of 0.25° (about 14 km at the equator, approximately the same size as the 13.5 km AIRS nadir footprint). AIRS is on a 1:30 pm equator ascending overpass orbit while ECMWF analysis/forecast are output at 3 hour intervals (8 per day) starting at 00.00 GMT. The closest in time forecast/analysis output is used to provide gridded fields, which are matched (using nearest grid point) to the AIRS L1B observations. This means the profile versus observed (latitude,longitudes) match ups are within 0.25 ±0.05 of each other, while the time differences are uniformly distributed within ±1.5 hours.

The topmost AIRS RTA pressure layer boundary is 0.005 mb, so US Standard temperature, water vapor and ozone fields (McClatchey et al., 1972) are appended above the 0.01 mb boundary. Standard profiles are also used for the remaining atmospheric gases, with carbon dioxide ($CO_2$) and methane ($CH_4$) concentrations set to 385 ppmv and 1.8 ppmv at the surface. Masuda ocean model emissivities (Masuda et al., 1988) are used, while land emissivities come from Zhou et al. (2011).

## 2.3 Radiative Transfer Models

The description of existing cloud representation and scattering codes for nadir infrared sounders include those found in Zhou et al. (2005); Liu et al. (2009); Chen et al. (2013); Ou et al. (2013); Vidot et al. (2015); Liuzzi et al. (2016), and Griessbach et al. (2013) for infrared limb sounders; separate examples can also be found for dust and volcanic ash aerosols.

We use two different RTAs, described below, to simulate AIRS infrared radiances that differ primarily in the scattering radiative transfer. Both RTAs use the same AIRS 100 pressure layer scheme (Strow et al., 2003); layer thicknesses range from 0.25 km at the surface, 0.75 km at the upper tropopause, and about 4 km at 0.005 mb (about 80 km) which is the TOA for the model.

### 2.3.1 SARTA

The clear sky version (with gray cloud capability) of SARTA is used for the NASA AIRS Level 2 retrievals. Layer optical depths are generated using pre-computed predictor coefficients (Aumann and Pagano, 2002; Strow et al., 2003, 2006). SARTA is trained using optical depths from the pseudo line-by-line (LBL) kCompressed Atmospheric Radiative Transfer Algorithm (kCARTA) package (De Souza-Machado et al., 2002). SARTA has been validated during dedicated AIRS campaigns (Strow et al., 2006).



We extended SARTA to handle clouds and aerosols, based on Parametrization of Clouds for Long-Wave Scattering in Atmospheric Models (PCLSAM) (Chou et al., 1999) algorithm. The PCLSAM algorithm recasts the extinction, single scattering albedo and asymmetry factor due to clouds and aerosols, into an effective absorption optical depth, and is used in other infrared fast models, see for example Matricardi (2005); Vidot et al. (2015); Liuzzi et al. (2016). For each SARTA AIRS layer that contains a cloud/aerosol, the total optical depth is then the sum of the atmospheric gas optical depth plus the cloud/aerosol effective optical depth. Fast, efficient clear sky radiative transfer can then be used to compute the TOA radiance, and to compute finite difference jacobians. Cirrus cloud scattering parameters come from Baum et al. (2007, 2011), while water cloud scattering parameters come from Mie coefficients integrated over a gamma distribution of variance 0.1.

### 2.3.2 PCRTM

We bench-marked the SARTA TwoSlab model versus the radiance simulator based on the Principal Component Radiative Transfer Model (PCRTM) (Liu et al., 2006, 2009) with full accounting of the cloud sub-grid variability (Chen et al., 2013). PCRTM is a fast model that computes atmospheric optical depths based on the Line-by-Line Radiative Transfer Model (LBLRTM)(Liu et al., 2006). PCRTM uses an adding cloudy sky scheme (Liu et al., 2009), based on reflectance and transmittance trained using 32-stream Discrete Ordinates Radiative Transfer Program for a Multi-Layered Plane-Parallel Medium (DISORT) (Stamnes et al., 1988) to simulate the effects of ice and water clouds. Again, ice scattering coefficients come from Baum et al. (2011) while the refractive indices for water come from Segelstein (1981). Unlike conventional channel-based radiative transfer models which compute the radiance of each channel separately, the PCRTM calculates the scores (i.e. the coefficients) of pre-computed principal components (PCs) in the spectral domain, with the instrument spectral response function taken into account. The PC scores contain essential information about the radiances, and can be calculated by performing monochromatic radiative transfer calculations at a small number of frequencies. The spectral radiances are then computed by multiplying the PC scores with pre-computed PCs. With this approach, the PCRTM achieves both high accuracy and extremely fast computational and high storage efficiency (Liu et al., 2006; Chen et al., 2013). Chen et al. (2013) showed that the root-mean-square differences between the PCRTM and LBLRTM (a widely used line-by-line radiative transfer benchmark model (Clough et al., 2005)) are 0.67 K for the clear-sky case and 0.78 K for the overcast case, for a wavenumber range spanning 0-2000 cm$^{-1}$. Chen et al. (2013) implemented a radiance simulator using the PCRTM and taking cloud variability into account in the same way the ISSCP simulator does(Klein and Jakob, 1999).

## 3 Cloud Model Field Conversion

Here we describe the TwoSlab cloud representation and the MRO cloud models. The latter is used exclusively with PCRTM, and the former with SARTA except in Section 4.2 when both PCRTM and SARTA use the TwoSlab model for inter-RTA comparison purposes. The MRO model has been previously documented and is briefly summarized at the end of this section.





### 3.1 TwoSlab conversion

Our cloud representation scheme replaces the *N*-level NWP cloud vertical profiles by one or two randomly overlapping finite width slabs clouds. The NWP cloud liquid water and cloud ice water content (CLWC,CIWC) profiles (in g/g) are integrated to obtain the column loading of the clouds (in $g/m^2$), and also to determine the slab cloud top/bottom pressures. The CIWC,CLWC

profiles, cloud cover profiles and total cloud cover are used to determine the slab cloud fractions. Effective particle sizes then need to be assigned to the clouds.

Infrared sensors cannot see through optically thick clouds and are mostly sensitive to the emission from cloud upper boundary while emission throughout the cloud can contribute to the outgoing radiance for less optically thick clouds. The TwoSlab model is very flexible when placing the slabs, for example (a) at the weighted mean or centroid (C) of the cloud ice or cloud

liquid profile or (b) near the most prominent cloud profile peak (P), which is best for optically thin clouds.

In practice, the cloud content profile $CXWC(z)$ (where $X = I, W$ for ice or water cloud) is smoothed before construction of the two cloud slabs (ice and water). The NWP cloud profiles usually result in the code identifying one water and one ice cloud, though the CLWC/CIWC pairs could produce two liquid or two ice slab clouds, or just a single slab cloud.

Figure 1 shows two examples of slab cloud outputs. The left panel is the simpler case where the NWP cloud profile (in

blue) is singly peaked. The right panel shows a case where the profiles are much more complex, and the cyan (water cloud) slab is placed higher in the atmosphere closer to the peak of the weighting function. The integrated cloud amount $g/m^2$ is proportional to the width of the cyan/magenta bars.

Assuming one ice and one water cloud slab are produced, the cloud fractions are constrained as follows:

$$\text{TCC} = c_{\text{water}} + c_{\text{ice}} - c_{\text{overlap}} \tag{1}$$

where TCC is the NWP total cloud cover (TCC). From this we compute the individual clouds fractions and their overlap using the following criteria :

1. If there is only one cloud present, its cloud fraction is set to TCC.

2. For two ice or two water clouds, the fraction for the first cloud $c_1$ is set as $TCC \times R$ where $R$ is a random number between 0 and 1. Then check to see if TCC is less than a random number; if so set $c_{\text{overlap}}$ to be $c_1 \times RR$ where $RR$ is
also random, else the overlap fraction is set to 0. $c_2$ then follows from Eq. 1.

3. If there is one ice and one water cloud, the cloud fractions are set according to $c_{\text{water}}, c_{\text{ice}}$ (described below); after that $c_{\text{overlap}}$ is set using Eq. 1.

For the third case, the water cloud slab fraction comes from weighting the NWP cloud cover $(CC)$ profile using the cloud liquid water content profile $c_{\text{water}} = \sum \text{CLWC}(z) \times \text{CC}(z) / \sum \text{CLWC}(z)$. The ice cloud fraction $c_{\text{ice}}$ is similarly determined.

Water particle effective diameters are 20 $\mu$m plus a uniformly distributed random offset. The ice effective particle size are estimated from a temperature based parametrization by (Ou and Liou, 1995), where the NWP temperature profile is used to associate the ice cloud slab top pressure with a cloud top temperature.





**Figure 1.** Example of cloud vertical profiles, reduced to one or two slabs. The red and blue curves come from the NWP model, while the cyan and magenta are the resulting locations (and loadings) for the slabs.



## 3.2 Radiance computation

The channel all-sky radiance $r_i(\nu)$ is computed using four weighted radiance streams

$$
\begin{aligned}
r_i(\nu) = {} & f_{\mathrm{clr}} r_i^{\mathrm{clr}}(\nu) + c_{\mathrm{overlap}} r_i^{(12)}(\nu) \\
& + cx_1 r_i^{(1)}(\nu) + cx_2 r_i^{(2)}(\nu)
\end{aligned}
\tag{2}
$$

where $f_{\mathrm{clr}}$ is clear fraction, $cx_i, i = 1,2$ is the exclusive cloud type $i$ fraction and $c_{\mathrm{overlap}}$ is the cloud overlap between the two cloud types; the exclusive cloud fraction being related to the cloud fraction via the relationship $cx_i = c_i - c_{\mathrm{overlap}}$. The model currently exclusively uses ice or water clouds when computing the radiances $r_i^{(1)}(\nu), r_i^{(2)}(\nu)$ associated with the cloud types; $r_i^{\mathrm{clr}}(\nu)$ is the clear sky contribution while $r_i^{(12)}(\nu)$ is the radiance contribution from the cloud overlap. Since the atmospheric gas optical depth computation dominates the run time, computing four radiance streams is not a speed penalty, and the overall
average run time per profile is about double that for a single clear sky radiance computation.

## 3.3 Maximum Random Overlap Conversion

The MRO cloud processing for the PCRTM model is described in Chen et al. (2013), and will only be briefly summarized here. MRO converts the NWP water and ozone levels profiles to 100 layer profiles. For each layer, the cloud ice water content and cloud liquid water content mixing ratios are converted to a cloud optical depth. The optical depths at each layer are summed.
Layers above 440 mb are considered ice clouds, and layers in the lower atmosphere are assigned to water clouds (Rossow and Schiffer, 1983, 1991). The effective water diameter is set at 20 $\mu$m while the effective ice diameter is again temperature dependent, based on the parametrization in Ou and Liou (1995). The cloud cover profile $cc(z)$ is used to generate 50 sub-columns using MRO (Chen et al., 2013) for which one radiance is computed per sub-column; the final radiance is an average over these sub-columns.

## 4   Inter-Comparisons of SARTA and PCRTM

### 4.1   Clear-Sky Comparisons

An earlier inter-comparison of the SARTA and PCRTM clear sky models is presented in Saunders et al. (2007). In this sub-section we assess the more recent spectroscopy embedded in the SARTA and PCRTM codes, using ECMWF thermodynamic profiles and surface parameters to compare clear sky radiances computed from the models.
We use 1600 randomly chosen night time scenes observed by AIRS on 2009/03/01 for an inter-RTA clear sky simulation comparison. The locations span all climate zones over ocean and land, as well as all AIRS scan angles. Night time scenes are used to avoid non-local thermodynamic equilibrium (De Souza-Machado et al., 2007) and solar surface reflectivity during the daytime in the 4 $\mu$m shortwave region. Both of these effects are handled differently by SARTA and PCRTM and are not relevant to this paper.



Figure 2 shows the calculated BT biases between SARTA and PCRTM clear sky models along with AIRS noise levels. The top panel shows the mean differences, while the bottom shows the standard deviations. The mean bias between SARTA and PCRTM clear calculations is within AIRS noise levels at all channels, except in the methane region (1300 cm$^{-1}$) and some channels in the water vapor 6.7 $\mu$m region. This is due to differing methane and water vapor spectroscopy and continuum models in these two RTAs. In addition, PCRTM uses a density weighted layer temperature that may introduce differences. Overall, differences between SARTA and PCRTM effective BTs are typically within AIRS noise levels.

**Figure 2.** Spectral differences for clear sky calculations between SARTA and PCRTM for ocean night scenes. Also shown are the AIRS noise levels. Typical difference between PCRTM and SARTA are less than 0.25 K, except in the methane region (1300 cm$^{-1}$) where SARTA does not use line mixing, and in some of the strong water vapor lines (1400+ cm$^{-1}$).





## 4.2 All-Sky Comparisons for TwoSlab Clouds

Here we compare all-sky radiances computed using SARTA and PCRTM, but use the same TwoSlab cloud representation in both RTAs. This tests the differences in each RTA's underlying scattering algorithm by keeping the cloud representation the same in both. Thus this directly compares the relative accuracy of the PCLSAM scattering algorithm used in SARTA against the DISORT-based scattering used in PCRTM.

The PCLSAM algorithm approximations in SARTA are more accurate for absorptive clouds that are more likely in the mid-IR. However, the DISORT-based scattering in PCRTM is more accurate if the cloud representation is correct. In general it would be reasonable to expect the differences to increase with optical depth and/or cloud fraction. In addition, in the TIR the single scattering albedo of water clouds is generally larger than that of ice, so we would also expect larger differences for water clouds.

To evaluate the SARTA (PCLSAM) versus PCRTM (DISORT) radiance differences, we used 1000 scenes spanning all climate types from AIRS on 2009/03/01. After matching the thermodynamic and cloud NWP fields to the observations, and subsequent conversion of the input cloud profiles to slab clouds, SARTA and PCRTM were run twice: (a) A clear sky run where no cloud effects are included, and (b) A all-sky run using the TwoSlab cloud representation derived from ECMWF.

In the TIR window region cloud forcing (difference between observed BT and surface temperature) can be as large as 100 K (for the Deep Convective Cloud (DCC) cases). For our sample set the mean (AIRS observation-SARTA TwoSlab RTA simulation) difference at 820 cm$^{-1}$ is -1.8 K compared to a mean cloud effect of 27.9 K. Similarly at 1231 cm$^{-1}$ the mean difference is -2.4 K compared to a mean cloud effect of 28.7 K. The corresponding biases for the PCRTM TwoSLab simulations are -2.4 K at 820 cm$^{-1}$ and -2.6 K at 1231 cm$^{-1}$. The effects of the clouds becomes less noticeable for channels sensing high in the atmosphere, such as the 650-700 cm$^{-1}$ and 1400-1600 cm$^{-1}$ regions. These comparisons shows the differences between the scattering RTAs is much smaller than the mean cloud effect.

Figure 3 shows the mean and standard deviations between SARTA TwoSlab/PCRTM TwoSlab using double differences, where the mean of the clear sky differences is removed from the mean of the all-sky differences. The double difference removes residual spectroscopic differences, allowing one to attribute the remaining differences to the RTA scattering algorithms. Differences are seen in the window where cloud scattering is most significant, but overall they are less than 0.5K. A detailed examination showed that the differences for ice clouds was proportional to the ice cloud fraction, while there was comparatively more scatter in the differences between PCLSAM and DISORT for water clouds at any cloud fraction.

These comparisons indicate that our implementation of the PCLSAM model is a fast, yet simple and effective method to include scattering effects in the TIR, as has also been shown by Matricardi (2005); Vidot et al. (2015); Liuzzi et al. (2016).

## 5 All-Sky Comparisons: TwoSlab versus MRO

We now compare radiances produced using the TwoSlab cloud representation model using SARTA and those produced using the MRO cloud representation using PCRTM, comparing both to AIRS all-sky observations. The AIRS data obtained on 2011/03/11 is used in this section, with co-located thermodynamic and cloud fields from the ECMWF model. The SARTA/TwoSlab







**Figure 3.** Biases and standard deviations between 1000 TwoSlab computations using SARTA and PCRTM. See the text for details on these double-difference signals.



calculations used slab clouds at the weighted mean ((C)entroid) of the cloud profiles. The 1231 cm$^{-1}$ channel is used to compare the observed versus computed radiances since it is largely free of atmospheric absorption except for several degrees of water vapor forcing in the tropics near the surface, but at the same time is strongly impacted by clouds. An important factor in the comparisons to actual AIRS observations is the ±1.5 hour mismatch between NWP forecast output and AIRS observations at

1:30 am/pm equator crossing time, which implies the cloud locations in the model are likely to be in the wrong position.

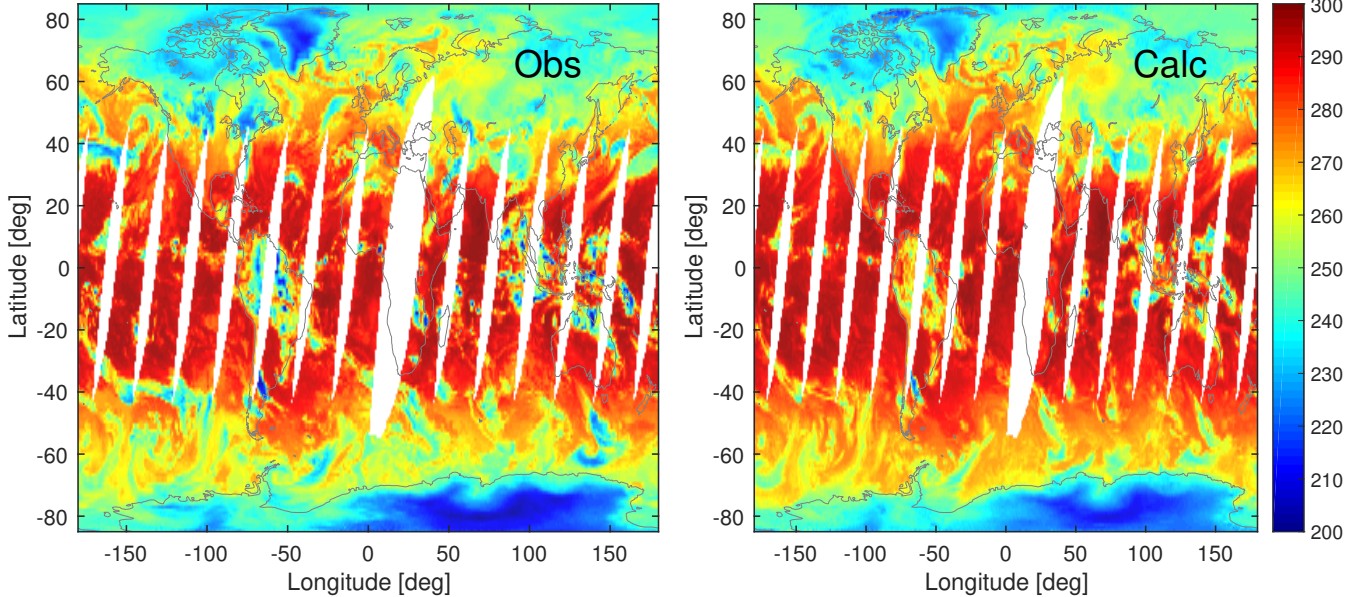

**Figure 4.** Left: AIRS 1231 cm$^{-1}$ channel brightness temperatures (in K) for the descending node (night) on 2011/03/11, using a 1 degree grid. Right: Computed BT 1231 using SARTA TwoSlab. Note the excellent agreement between the observations and calculations - there are large areas in the tropics with ~0 K differences. Careful examination (of obs-cal) does reveal large negative and positive biases scattered throughout due to spatial mis-matches in the model versus observed clouds, for example the Tropical West Pacific (see Figure 5) and the Amazon.

The left panel of Figure 4 shows the 1231 cm$^{-1}$ BTs for the night time overpasses on 2011/03/11. The data are averaged over a 1 degree grid for plotting purposes. The white areas are gaps between the ~2000 km wide AIRS swaths. The right panel shows the BT 1231 cm$^{-1}$ calculated with SARTA TwoSlab. The observations in the left panel show that the BTs vary from 330K (hot surface, no clouds) to as low as 210K for deep convective clouds in the tropics. Over tropical oceans, most of the

pixels in the right hand panel agree very well with the observations, indicating almost zero difference between observations and calculations generally in regions with few clouds. Conversely the Tropical Warm Pool (TWP) and the Amazon show spatial differences, due to small temporal and spatial errors in the model, especially for Deep Convective Clouds (DCC) which have extremely cold cloud tops; in particular note the almost total lack of DCC over the Amazon compared to the observations. However, these results show that overall the dynamical distribution of moisture and clouds is well represented by ECMWF as

noted by Allan et al. (2005) and (Shahabadi et al., 2016).





Figure 5 is a zoom of the Tropical Warm Pool (TWP) region. Extremely cold cloud tops are clearly seen by AIRS (left panel). The right panel plots the differences between observed and computed all-sky radiances, clearly showing unsurprising offsets between observed and computed convective structures, which could be due to both model errors and to the ±1 1/2 hour time offset between observations and model fields. The rapid varying spatial differences in these biases of opposite sign suggests

that the model clouds are relatively accurate, they just have slightly different spatial patterns.

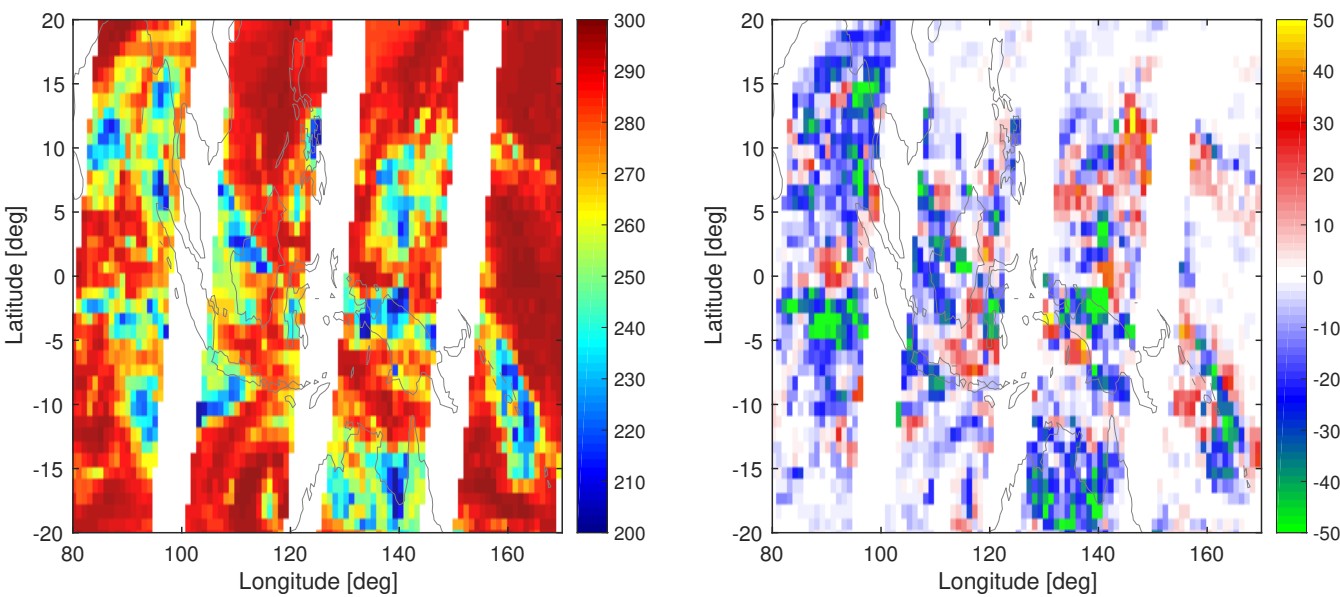

**Figure 5.** Left: Zoom of left panel in Figure 4 in the Tropical West Pacific region. Right: Zoom of the model biases highlighting that most large biases are due to small spatial offsets of the model clouds. Note that red/blue pixel are often adjacent to each other. For the region shown, the mean bias and standard deviation of (observations - SARTA TwoSlab) is -6.1 ± 20.1 K.

Plots of the 1231 cm$^{-1}$ all-sky calculations using PCRTM/MRO are very similar to Figures 4 and 5, namely large areas of the tropical oceans having almost zero bias, and with noticeable mis-matches of cold cloud tops.

## 5.1 Window Channel PDFs

Here we explore the similarities and differences between the observations, SARTA/TwoSlab, and PCRTM/MRO by examining

the radiance Probability Distribution Functions (PDF)s and the scene dependence of the mean BT differences, again for the 1231 cm$^{-1}$ AIRS channel.

Cloud mis-match errors will contribute to a significant portion of the standard deviation between observations and calculations, as is evident in from Figures 4 and 5. Conversely as mentioned earlier, the dynamic range spanning 200K to 330K is seen both in observations and calculations from both cloud representation models. The left panel of Figure 6 is a plot of

the corresponding histograms or un-normalized probability distribution functions (PDFs); the bins are 1 K wide. The curves are the night time observations (black), SARTA/TwoSlab calculations (blue/cyan) and PCRTM/MRO calculations (red). The





blue and cyan SARTA TwoSlab calculations differ in the positioning of the slabs: the (P)eak of cloud weighting function or the (C)entroid of the cloud profile respectively, as described in Section 3.1. The calculated radiance histograms are very similar compared to their differences with the observations. If the 1231 cm$^{-1}$ PDFs are subset for different geographical regions discrepancies between the computed and observed PDFs are easier to evaluate compared to the global PDFs. For example:

1. In the tropics the observations have more cold (DCC) scenes, also seen in (Shahabadi et al., 2016).

2. The frozen oceans in the northern polar regions and off the Antarctic coast have BT1231 calculations between 240-260 K, which are quite different than what is observed.

3. A significant portion of the BT1231 calculations between 260 and 280 K come from the extra-tropical oceans (-40 S to -70 S and +40 to +60 N)

4. The whisker plots (circles are means while the bars are the standard deviation) show the calculations are generally slightly warmer than the observations, while the spreads are all very similar.

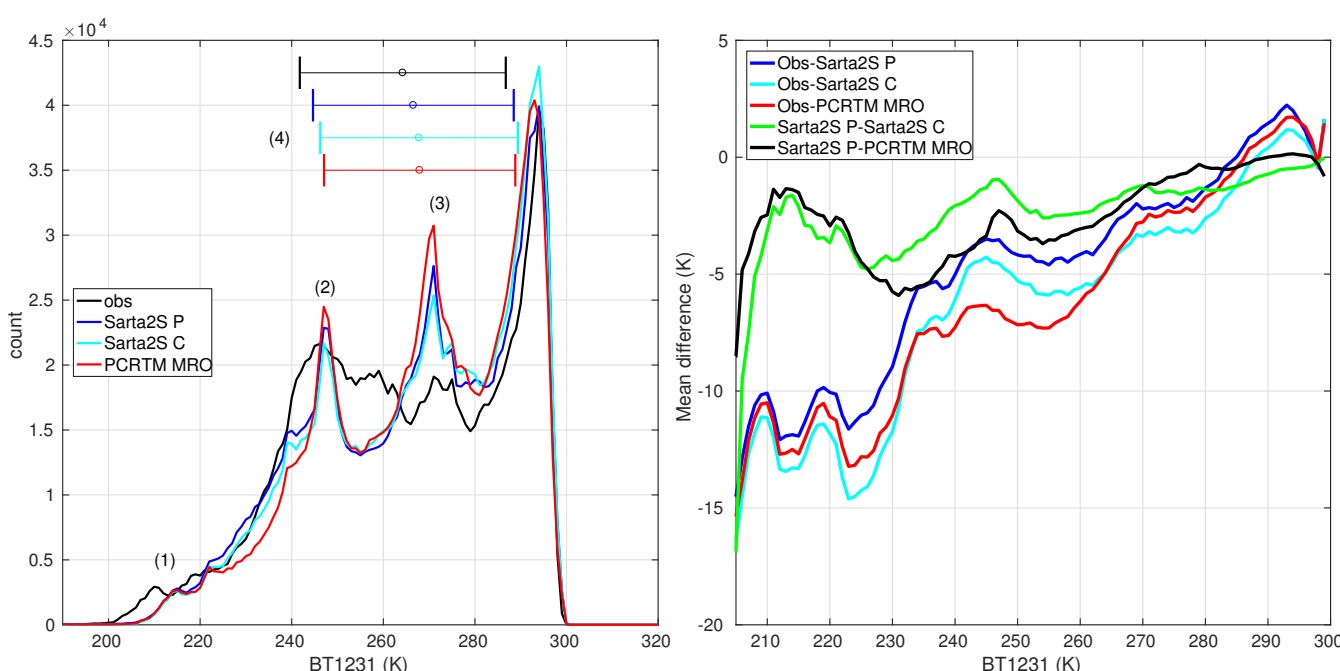

**Figure 6.** Left: 1231 cm$^{-1}$ channel brightness temperatures PDFs (probability distribution functions) for night scenes (global). Locations (1),(2) and (3) highlight large differences that are discussed in the text, while the horizontal lines and associated circles in Location (4) represent the BT1231 cm$^{-1}$ means and standard deviations. Right: Mean biases as a function of scene temperature for various RTAs tested here. Typical standard deviations between the different RTAs is about 10K, while the typical standard deviations between observations and calculations vary from 30K (cold scenes) to 10 K (warm scenes).





The right panel of Figure 6 plots the mean difference between various pairs of observations and calculations as a function of the scene radiative temperature, BT1231. The abscissa is constrained to be 205 K to 300 K, as the low number of observations outside these limits yields large average differences. Positions (1),(2) and (3) are as for the left panel. Typical standard deviations are ±10 K for the blue, cyan and red (obs-calc) curves, and ± 5 K for the green and black (inter-model) curves. The

calculations agree well with observations for high BT1231 (280-300 K) values where clouds are less important. As the scene temperature lowers the observations indicate a lack of high clouds in the model, which has been noted previously (see for example (Shahabadi et al., 2016)). The differences between SARTA (P/C) and PCRTM-MRO are generally small compared to the observation versus model differences.

Note that for the highest clouds SARTA (C) is hotter than SARTA (P), which makes sense since these are very optically

thick clouds and the TIR weighting function peaks at the cloud top. For warm scenes, the MRO simulations are again closer to SARTA (P) than SARTA (C) indicating that, as expected, placing clouds near the weighting function peak in the TwoSlab algorithm is preferable to the centroid.

Table 1 summarizes the 1231 cm$^{-1}$ window channel global statistical comparisons including differences with observations and differences among the various RTAs. First note that the high standard deviations for observations minus computed radiances

are dominated by spatial/time mis-matches, and are not necessarily indicative of model limitations. For the purposes of this paper, the most interesting result is that the SARTA (P) agrees better with observations in the mean, while SARTA (C) generally agrees better with MRO. The uncertainties in the model cloud fields are sufficiently large (especially given spatial/time mis-matches) that these comparisons are not sufficient to indicate which scattering model is more accurate.

**Table 1.** Night land and ocean 1231 cm$^{-1}$ biases for 2011/03/11 in K. "P" and "C" denote the cloud slabs placed at the (P)eak of the weighting function and the cloud profile (C)entroid respectively.

| Region | Stemp - Obs | Obs - PCRTM/MRO | Obs - SARTA/2S (P) | Obs - SARTA/2S (C) | MRO-2S(P) | MRO-2S(C) | 2S (P-C) |
|---|---|---|---|---|---|---|---|
| Global | 12.5 ± 14.3 | -3.69 ± 10.4 | -2.3 ± 11.0 | -3.6 ± 10.8 | 1.4 ± 4.9 | 0.1 ± 4.5 | -1.2 ± 4.5 |
| Tropical | 15.4 ± 17.9 | -3.1 ± 13.7 | -1.9 ± 14.1 | -4.0 ± 13.7 | 1.3 ± 5.3 | -1.0 ± 4.0 | -2.2 ± 4.0 |
| N. Midlat | 12.8 ± 13.6 | -3.9 ± 9.3 | -2.6 ± 9.7 | -3.8 ± 9.9 | 1.3 ± 4.6 | 0.1 ± 4.3 | -1.2 ± 4.3 |
| S. Midlat | 14.4 ± 12.7 | -4.2 ± 9.5 | -3.5 ± 9.9 | -5.4 ± 9.6 | 0.7 ± 4.4 | -1.2 ± 3.5 | -1.9 ± 3.5 |
| N. Polar | 6.9 ± 9.4 | -2.9 ± 7.1 | -1.2 ± 7.8 | -0.3 ± 8.1 | 1.7 ± 4.5 | 2.6 ± 5.2 | 0.9 ± 5.2 |
| S. Polar | 9.9 ± 9.2 | -4.9 ± 6.5 | -2.8 ± 8.0 | -3.5 ± 7.3 | 2.1 ± 5.0 | 1.4 ± 4.3 | -0.7 ± 4.3 |

We note the PDF correlations between observations and all the calculations, and also among the calculations themselves

is typically 0.9 and above, which reinforces the point that the NWP fields from ECMWF capture much of the atmospheric variability that is observed; however the mis-match between observed and model cloud tops lead to biases on the order of 2-4 K with standard deviations on the order of 10 K.

The implications of this are that by shifting the position of the model clouds, one could significantly mitigate the differences, and hence have *a-priori* micro-physical properties of the TwoSlab clouds, for a physical single footprint all-sky retrieval. This

idea is exploited later in the section on applying the TwoSlab code for use in single pixel all-sky retrievals.





We close this sub-section with Figure 7 where the BT1231 observations and calculations are plotted as pdfs for the tropics, mid-latitudes and polar regions. Note that we have limited the pdfs to contain data only over non-frozen oceans, using the ECMWF sea ice fraction model field. As in Figure 6 the black curves are observations, while SARTA/TwoSlab (P)eak and (C)entroid calculations are in (blue/cyan) and PCRTM/MRO calculations are in red. While the tropical PDFs are quite similar

between observations and both RTAs, the mid-latitudes suggest ECMWF produces more clouds than observed. The polar PDF calculations are again quite similar, but with even large disagreements with observations. Accounting for this in detail is not the focus of the paper, but some insight was gained by comparing the cloud fields in Granule 001 obtained over the Southern Ocean/Antarctica (SOA) against those in Granule 137 which was filled with Marine Boundary Layer (MBL) clouds off the coast of Namibia. For the SOA granule the mean CIWC and CLWC were $1 \times 10^{-5} g/g$ and $0.3 \times 10^{-5} g/g$ with peaks centered

at 825 ±50 mb and 875 ±50 mb respectively, and mean cloud fractions of 0.3 at about 800-900 mb, and less than 0.1 above that. Conversely the mean CIWC amounts for the MBL granule were almost 20 times larger at roughly the same vertical position (while the CIWC amounts were 10 times lower and cloud fractions were about the same); however the computed radiances were about 5-10 K cooler than the surface temperatures, in rough agreement with the observations. The mean surface pressure and temperature for the SOA granule was 985 mb and 273 K respectively, while the mean atmospheric temperature at 850 mb

was 262 K. The bottom-most panel of Figure 7 shows the observed peak was close to 260 K compared to the computed peak at about 270 K. All this evidence points to one of two possibilities about the polar over-ocean clouds in the NWP model inability to statistically reproduce the observations. They could either at too low an altitude, or at the right altitude but not optically thick enough or with low cloud fractions.

## 5.2  Spectral Comparisons

Finally we compare the observed and calculated spectra for ocean scenes where the ECMWF model sea surface temperature is very accurate and the emissivity is well known. As in Figure 7, we divide these observations into tropical, mid latitude and polar zones (using boundaries at ±30 and ±60), with over 200,000 observations per zone. Regions with ice contamination (according to ECMWF) were removed to avoid uncertainties in emissivity for the simulations. As with the PDF plots, the spatial mismatches mis-matches between modeled and actual clouds will still lead to significant differences between observations and

calculations, as seen in from Figure 4.

These comparisons are summarized in Figure 8. In each panel the blue, green and red refer to the tropical, mid-latitude and polar regions; the solid lines (marked (S)) refer to the SARTA/TwoSlab calculations, while the lines with 'X' (marked (P)) refer to PCRTM/MRO calculations. Note in this figure the label (P) refers to PCRTM/MRO, and the SARTA calculations all use peak cloud weighting.

The mean tropical SARTA/TwoSlab and PCRTM/MRO calculations are slightly warmer than the observations, partially due to fewer/warmer deep-convective clouds in the ECMWF model. The mid-latitude window region calculations are on average about 5K warmer than the observations. Similarly in the polar regions the calculations are also warmer than the polar observations, though the SARTA/TwoSlab and PCRTM/MRO clouds simulations are closer to each other than to the observations.


**Figure 7.** Brightness temperature PDFS for the night non-frozen ocean 1231 cm$^{-1}$ channel for the data used in Figure 8 separated tropical, mid-latitude and polar regions. The whiskers correspond to the brightness temperature means and standard deviations.





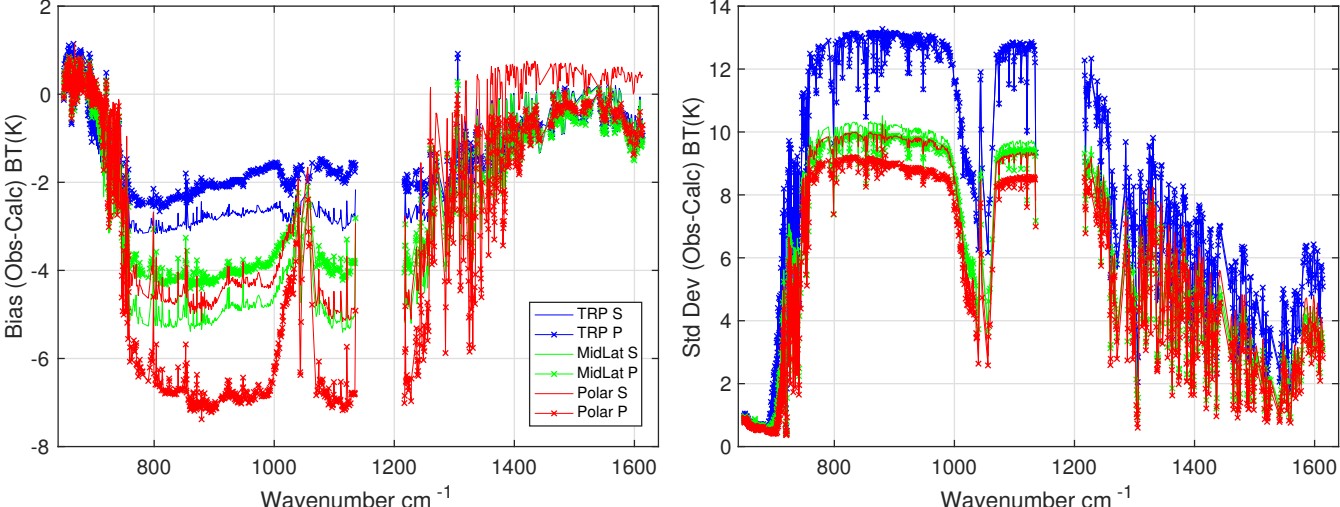

**Figure 8.** Mean biases (left) and standard deviations (right) between observations and calculations for ocean night scenes on 2011/03/11 for tropical (TRP), mid-latitude (MidLat) and polar (Polar) regions. Curves with ⋆ markers are for (P)CRTM/MRO. Solid lines denote (S)ARTA/TwoSlab PeakSlab. Results are similar in the tropics and mid-latitudes, with larger differences at the poles. See Figure 7 for the corresponding BT1231 pdf plot.

As expected from studying the 1231 cm⁻¹ PDFs, the largest differences between the observations and calculations are in the polar (red) region. The tropical biases were typically the smallest, though their standard deviation is the largest (as the models and calculations have to span warm surface temperatures all the way to cold DCC cloud tops). The largest spectral biases are in the window regions, which is to be expected as cloud effects are most readily seen in these regions.

5 The above plots are similar to all-sky monthly global averaged biases seen in (Shahabadi et al., 2016). They are put into context by considering the cloud forcing effect, which we simply define as the mean BT1231 difference between clear sky calculations and observations. In the tropics/mid-latitudes/polar regions they are 7.4 K, 10.2 K and 12.6 K respectively. Conversely the PCRTM/MRO versus SARTA/TwoSlab differences are a factor of 10 smaller, at -0.78 K, -0.78 K and 1.89 K respectively. Again our main emphasis here is to validate the accuracy of our simple SARTA TwoSlab algorithm relative to the 10 more rigorous PCRTM-MRO RTA, not to evaluate the accuracy of the ECMWF model clouds.

## 6 Application of TwoSlab Code : All-sky Retrieval

One full AIRS granule (6 minutes, 12150 spectra) worth of all-sky retrievals using the SARTA/TwoSlab code are provided here using the OEM method, which provides natural diagnostics such as DOF and AK that are extremely helpful in understanding the information content of our retrieval approach in the presence of highly variable clouds. This is a "proof of concept" retrieval 15 which has been separately tested on several days worth of AIRS observations. While considerable efforts have been put into selection of various regularization matrices and channels, we anticipate additional fine-tuning in the future.



## 6.1 OEM Approach

We follow normal OEM notation here, where the observation vector $y(\nu)$ (brightness temperature) is modeled by the radiative forward model operator $F$ and $\epsilon(\nu)$ is the combined instrument and forward model noise,

$$y(\nu) = F(x, \nu) + \epsilon(\nu) \tag{3}$$

5 where $\nu$ is the wavenumber, and $x$ is the thermodynamic and cloud state. The solution is regularized with matrix $R$ using a cost function $J$ (Rodgers, 2000; Steck, 2001) given by

$$\begin{aligned} J &= (y - F(x))^T S_\epsilon^{-1}(y - F(x)) \\ &\quad + (x - x_a)^T R(x)^{-1}(x - x_a) + J_{\text{sat}} \end{aligned} \tag{4}$$

The first two terms are the observation and background penalties respectively, while the last is a constraint to reduce the amount 10 of humidity supersaturation (Phalippou, 1996; Deblonde and English, 2003). Our regularization matrix contains both empirical regularization (Tikonov) and *a-priori* covariance-based terms. $J$ is minimized using a nonlinear Gauss-Newton iterative approach. (Rodgers, 2000; Steck, 2001)

$$\begin{aligned} x_{n+1} &= x_n + (K^T S_\epsilon^{-1} K + R^{-1})^{-1}(K^T S_\epsilon^{-1}(y - F(x_n)) \\ &\quad - R^{-1}(x_n - x_a) - J'_{\text{sat}}(x_n)) \end{aligned} \tag{5}$$

15 where $K$ is the jacobian. At present the observations covariance matrix $S_\epsilon^{-1}$ is diagonal, combining a linear sum of forward model error ($\leq 0.2$ K per channel) and AIRS channel dependent noise error. The retrieval currently uses no AIRS shortwave channels (past 2000 cm$^{-1}$) but uses almost the same 500 longwave channels used in the AIRS L2 retrieval.

### 6.1.1 State vectors and OEM parameters

The state vector consists of surface and profile temperatures (Kelvin), the logarithm of the water vapor profile (molecules/$cm^2$), 20 as well as a (logarithmic) multiplier for the ozone profile and two slab cloud amounts. The jacobian matrix $K$ in Equation 5 is built using these parameters. For any FOV the iterations were halted after $i_{\max} = 5$ iterations, or if there was no improvement in the $\chi^2$ after iteration $i \leq i_{\max}$.

The regularization matrix $R$ is block diagonal for the temperature and water vapor profiles, and is a linear combination of altitude dependent covariance matrices (with exponential decaying off-diagonal elements) and a $L_1$ type first-derivative 25 Tihkonov smoother. Diagonal terms were added for the remaining state variables being retrieved. The humidity saturation penalty is of the form $J_{\text{sat}} = \sum_{i=1}^{i=N} J_i$, where $J_i$ is set to 0 if the relative humidity (RH) of the $i$-th layer is less than 100 %, else it is computed from $r(log_{10} \frac{RH(i)}{100})^3$ where $r = 100.0$.

For this paper we start with smoothed climatological profile, and use 2 K uncertainties for the temperature and surface temperature, 60% uncertainty for water vapor profiles and 10% cloud loading uncertainty. We start with ECMWF surface 30 temperatures since they are likely to be better than climatology. Land and ocean surface emissivity is set from a database (see Masuda et al. (1988); Zhou et al. (2011)).



### 6.1.2 Retrieval *a-priori*

The highly non-linear effect of clouds on infrared radiative transfer makes retrieval success highly dependent on an accurate linearization point for cloud parameters. This fact has made it difficult to create physically-gased (not statistical) robust single-footprint infrared hyperspectral retrievals. Since typical infrared all-sky retrievals only have 2-4 degrees of freedom for cloud

fields (see Appendix 10) it is essential that the linearization point and a-priori for clouds be as accurate as possible. Fortunately, NWP model forecasts such as from ECMWF provide reasonably accurate cloud fields derived using the best physics possible for an operational model. However, as we have shown earlier, perfect spatial placement of model clouds is not possible, and winds can move forecast clouds significantly during the ±1.5 hour time difference between observations and an ECMWF analysis/forecast file.

The ECMWF cloud fields are statistically quite accurate in the sense that they reproduce similar spatial distributions of window channel brightness temperatures as the AIRS observations. Our approach is to compare the observed window channel brightness temperatures to those we compute from the ECMWF model (and cloud) fields using our TwoSlab approximation. We then find the closest spatial match between a particular window region observation and nearby simulated window channel radiance (in a least squares sense). The cloud fields from the ECMWF grid point with the closest matching radiance are then

used as the linearization point for the retrieval for the AIRS scene. We only retrieve cloud loading after the linearization, while keeping particle effective size, cloud top and bottom pressure and cloud fractions unchanged. The temperature and water vapor profile linearization point and *a-priori* is **not** taken from ECMWF, instead a monthly climatology is used.

For example, Figure 5 shows the overall spatial agreement between observed and simulated clouds over the Tropical West Pacific, the ECMWF cloud fields are often offset from the observations (and are a factor of two fewer). The 300 hPa ECMWF

(u,v) wind fields for this granule suggests that these convective regions are moving westward at approximately 36 km/h. Since the forecast is approximately 0.8 hours previous to the AIRS overpass, the model clouds could move by up to 30 km after the model forecast, or roughly two AIRS FOVs. Thus, just the time delays between forecast and observations can contribute to the cloud mis-match.

The temperature and water vapor profile *a-priori* state for the tests reported here used using co-located 10-year monthly aver-

aged AIRS L3 fields (March 2004-2013). The water vapor *a-priori* uncertainty was set to 60% and the temperature uncertainty to 2 K while the surface temperature used a 0.3 K uncertainty.

We chose a climatology for the *a-priori* in order to more easily demonstrate the performance of the retrieval algorithm. If a single-footprint retrieval was used for production of a long time series of AIRS Level 2 products we believe a reanalysis (ECMWF, MERRA) would be a more suitable *a-priori* in that it would provide accurate profile estimates below thick clouds.

The OEM framework will naturally provide this capability since the degrees-of-freedom of the retrieval shrink rapidly as the cloud thickness increases.





**Figure 9.** 1231 cm$^{-1}$ observed brightness temperatures for Granule 039 on March 11, 2011.





**Figure 10.** Three cross-sections corresponding to the black lines in the granule image are shown. The curves are BT1231 cm$^{-1}$ comparisons between the observations (black) and ECMWF (blue), which show the hit or miss characteristics of the spatial placement of the model clouds. The red curve just shows that we can find and include nearby clouds in the ECMWF model that have similar brightness temperatures to what is observed.



## 6.2 Single Granule Case Study

This section focuses on AIRS granule 039 from March 11, 2011, a day scene over the Tropical Western Pacific containing many DCCs. Figure 9 maps the 1231 cm$^{-1}$ BT observations. The three lines marked (A), (B) and (C) are at AIRS scan angles of roughly -23, 0 and +24°. Figure 10 shows the observations (black), retrievals (red) and original ECMWF calculations (blue)

for these three lines. The significant offsets in the original (ECMWF) clouds is apparent by comparing the blue and black curves. The red curve shows how well the final retrieval simulates the window temperatures, due to a combination of moving the ECMWF clouds fields to the appropriate pixels together with the adjustment of the cloud fields during the retrieval. The final calculations largely reproduce the observations. A more thorough examination of (longitude,latitude) versus BT1231 scatter plots after the retrieval show that the spatial patterns after the retrieval, including the cold DCC, correlate extremely

well to the patterns seen in Figure 9.

### 6.2.1 Spectral Biases and DOF

A first step in testing retrieval performance is to examine the final retrieval residuals and their standard deviations shown in Figure 11. Also shown are the initial differences between all-sky radiances simulated with ECMWF (clouds in their original positions) and the AIRS observations (blue curve). Most of the bias relative to ECMWF is due to clouds and some significant

water vapor differences. Our retrieval BT bias and standard deviation are given by the red curve. (The standard deviations from the original ECMWF co-located cloud fields have been multiplied by 0.2 to fit on the graph.) Most channel biases are in the 0.1-0.2K range although larger biases are evident in the 650-700 cm$^{-1}$ stratospheric region. Further work is needed to determine the cause of these deviations. We have included all retrievals here, including those with low degrees of freedom where the *a-priori* is weighted heavily.

The retrieval standard deviations are far smaller than those for ECMWF in the region of strong water lines past 1400 cm$^{-1}$. In this region cloud cover is unimportant except for the deepest DCCs, so this reduction in the standard deviations indicates skill in the retrievals. This is discussed in more detail below.

Figure 12 maps retrieval DOFs for this granule. Comparisons to Figure 9 show the clear correlation between low window channel BT, due to clouds, and low retrieval DOFs. The DOFs range from 5 for thick DCC (ice cloud loadings of 100+ g/m2)

and increase to larger than 10 for nearly clear scenes. These DOF values are more than a factor of two smaller than what is reported in Appendix B, where a diagonal-only error covariance matrix is used when assessing the *N*-level cloud information content. The red circles in the figure are the 60% of the observations in this granule where the L2 retrieval failed all the way down to the surface.

### 6.2.2 Thermodynamic retrievals

Temperature and water vapor retrieval statistics are shown in Figures 13 and 14. Statistical measures for a single granule, especially in the tropics, are of limited value for understanding retrieval accuracy, especially for temperature. However, they do help indicate nominal performance and provide a measure of the impact of clouds on the DOFs and retrieval accuracy. In





**Figure 11.** Retrieval results for Granule 039 shown in Figure 9. The original biases between observations and ECMWF before the retrieval and cloud replacement are also shown (in blue), note that this standard deviation has been reduced by 5X for plotting.





**Figure 12.** Retrieved degrees of freedom for G039 show evident dependence on observed BT1231 (which depends on cloud loading). The red circles denote locations of the AIRS FORs (3x3) where the retrieval quality down to the surface is 2, meaning missing or do not use; this happened for 60% of the L2 FORs.



addition, although ECMWF is quite accurate globally, it is difficult to judge how well ECMWF represents truth for a single granule but it is likely far better than our *a-priori* climatology.

These statistics have been separated by the total number of DOFs in the retrieval. Figure 13 shows scenes with DOFs less than 6.5 (thick cloud, 76 FOVs), and while Figure 14] have DOFs greater than 9 (almost or completely clear, 8165 FOVs). The small number of scenes with low DOFs also limit the utility of their statistics. The ECMWF and climatology profiles were multiplied by the retrieval averaging kernels for these comparisons.

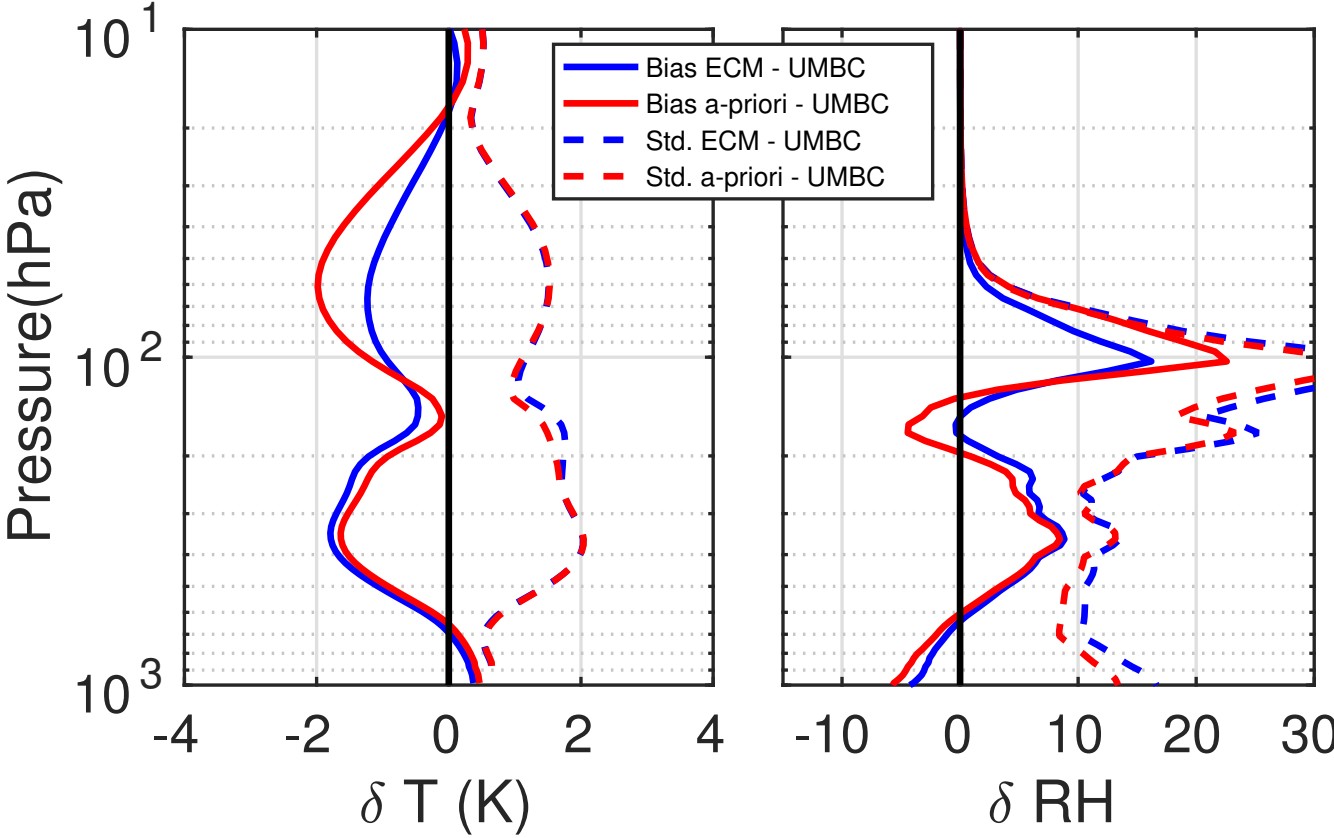

**Figure 13.** Retrieval statistics for Granule 039 for thick clouds, defined by the number of DOFs to be < 7. Comparisons between ECMWF profiles (multiplied by averaging kernels) and (blue) our retrievals (red) starting climatology. standard deviation. Note that the UMBC retrieval used over 95% of scenes in the granule while AIRS L2 had a 60% yield of good/best QA down to the surface; the interesection here is 77 profiles. Very similar plots are obtained if we use all 1592 low DOF retrievals.

The low DOF case (Figure 13) shows little difference between retrieved profile temperature and *a-priori* in the lower troposphere. This is not too surprising for a tropical granule. The cloud contamination that caused these low DOFs led the retrieval to stick to the *a-priori* in the troposphere. Small movements in the upper-troposphere and stratosphere did occur that gave




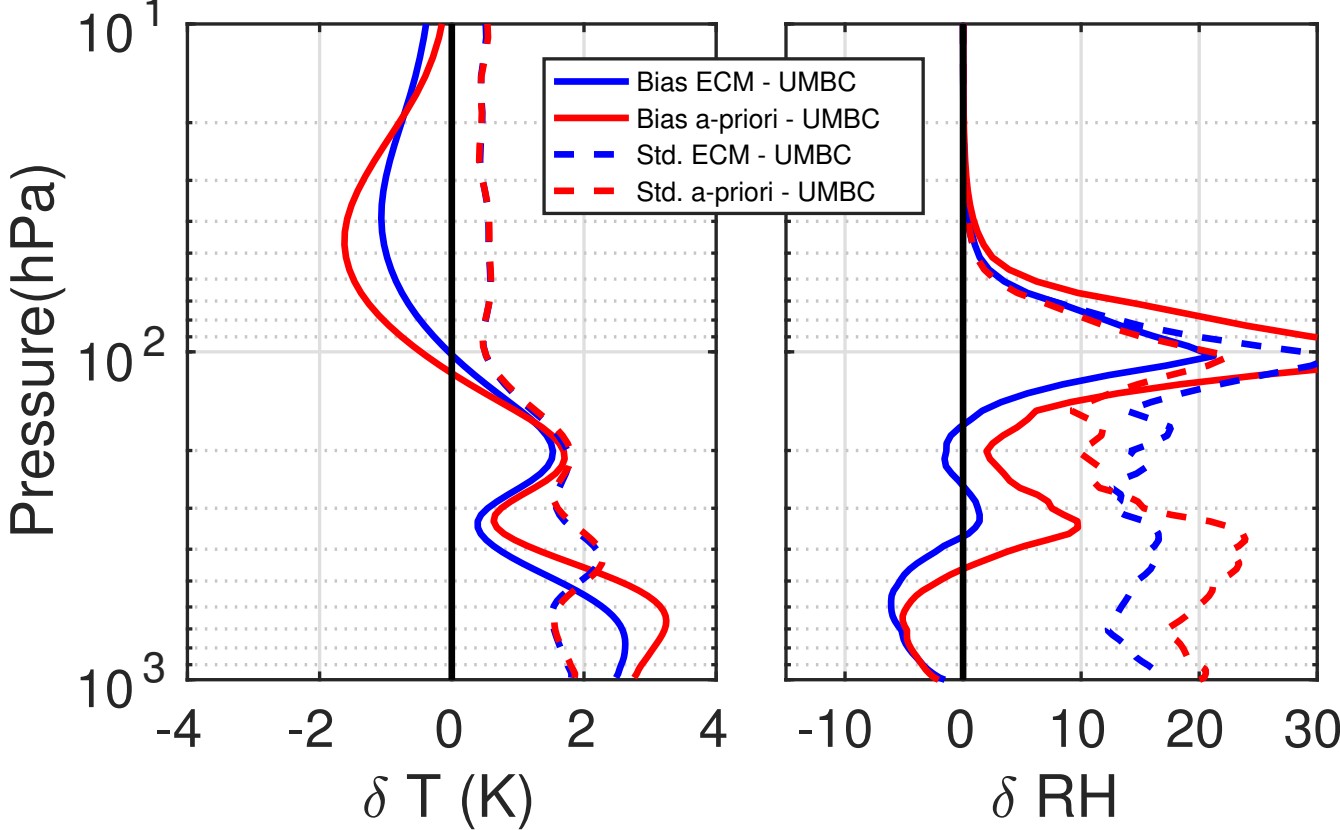

**Figure 14.** Same as Figure 13 except now uses cases for thin clouds, defined by DOFS to be > 10; 6220 profiles were used here (compared to 7296 if we ignore the AIRS L2 QA and only use high DOF retrievals, again with very similar plots being obtained).

similar disagreements with the *a-priori* and ECMWF. The same is true for water vapor for the low DOF case, any differences between the retrieval and either the *a-priori* or ECMWF are in the upper troposphere, where the cloud contamination is lower.

The high DOF cases in Figure 14 indicate that the retrieval moved slightly away from the *a-priori* towards ECMWF at almost all levels. Results are a bit more mixed for water vapor. The retrieval standard deviation from ECMWF is lower than

5 with the *a-priori* in most of the troposphere.

A much more definitive diagnostic of retrieval performance is given in Figure 15, which is a curtain plot of relative humidity along the line denoted by "B" in Figure 9. Here we compare a series (from top to bottom) of our retrieval, ECMWF, AIRS L2 retrieval and *a-priori* humidity profiles, with the vertical axis being pressure and horizontal axis being latitude. The bottom panel is a plot of the BT 1231 cm[-1] channel which is an excellent proxy for cloud top height and opacity. The *a-priori* and

10 ECMWF water vapor structures have been multiplied by the averaging kernels.

There is little water vapor structure in the *a-priori* (fourth panel) given that is a monthly average of many years. The retrieved water vapor (top panel) shows significant structure along this track, with some small instabilities in the regions of thick high





**Figure 15.** Comparing Relative Humidity (colorbar) along track "B" in Figure 9. From top to bottom we have our retrieval, ECMWF, AIRS L2 retrieval and *a-priori* humidity profiles. The bottom panel is a plot of the observed BT 1231 cm$^{-1}$ channel. The black and red circles in the top panel are the positions of ice and water clouds, with the circle size denoting the OD.





clouds (near -12 deg latitude and +6 degrees latitude); the black and red circles mark the positions of ice and water clouds, with the circle size representing the optical depth. The ECMWF water fields show very significant structure, which becomes quantitatively similar to our retrievals once the retrieval averaging kernels are applied to the ECMWF profiles. Overall the retrieved water vapor fields move from being almost structureless to showing similarities to those of ECMWF, indicating

very encouraging retrieval performance. For example both are relatively dry at 500 mb between -5°S and the equator, at approximately the same locations where AIRS L2 were drier. Overall our retrieval is showing higher humidity values than either L2 or ECMWF. In particular where the bottom panel shows medium clouds (for example at -11°S, -3°S and +6°N) our retrieval shows high humidity at 400-600 mb, and where the bottom panel shows thick high cloud (for example between 0°and +3°N) the retrieval returns the *a-priori*.

The ability of the retrieval to catch some of the upper-troposphere variability near 150 hPa seen in ECMWF indicates good vertical resolution as well. Note that the retrieval does not use any information from ECMWF, except for the clouds. We also comment that the UTLS humidity from the AIRS L2 is significantly lower than either ECMWF or our retrieval, with the blanked out areas indicating where the surface AIRS L2 QA flags were bad.

The higher tropopause RH that was initiated by climatology remained unaffected by the retrieval; this could be alleviated by

an improved first guess of the thermodynamic state, as well as choosing WV channels that peak very high in the atmosphere. In the future we plan to use reanalysis as *a-priori*, which will be adjusted by the retrieval when there are low to medium optically thick clouds. The use of a fixed shape for the ozone profile is also a limitation of the present retrieval that will be removed in later versions.

### 6.2.3   Cloud Parameter Changes

Comparisons between the initial EMCWF TwoSlab cloud parameters (found by matching window BTs to nearby ECMWF scenes) and the retrieved cloud parameters have a number of understandable differences. It is well known that NWP models do not produce as many deep convective clouds as observed so it is understandable that the mean ice cloud fraction changed from less than 0.5 to higher values ranging from 0.6 to 0.9 (these can be quite thin ice clouds). The water cloud fractions increased slightly from when less than 0.3, and generally decreased for the higher factions. In addition, the frequency of high ice cloud

tops (less than 250 mb) increased while the rest decreased; for water clouds the largest increase in frequency of occurrence was seen between 500-700 mb.

A quick validation of our ice cloud optical depths is achieved by comparing of AIRS L2 ice optical depths versus our retrieved ice cloud loading (in g/m2). While the thermodynamic AIRS profiles are generated at 45 km (3x3 AIRS footprint) resolution, the ice cloud AIRS L2 retrievals are generated for single footprints (Kahn et al., 2014) after the L2 thermodynamic

retrievals, in a separate step that keeps all other retrieval variables constant. Figure 16 is a comparison of AIRS L2 ice optical depths versus our retrieved ice cloud loading (in g/m2), clearly showing a proportional relationship. This is very encouraging in that we are also retrieving the full thermodynamic state and water cloud parameters simultaneously.

The retrieval used cloud heights derived from matching to nearby ECMWF cloud fields. Figure 17 provides partial validation of that approach by comparing Moderate Resolution Imaging Spectrometer (MODIS) L2 ice-cloud heights with the initial







**Figure 16.** Retrieved ice cloud amount (in g/m2), compared to AIRS L2 ice cloud optical depths. The colorbar is the log(10) of the number of points in the bin.





ECMWF ice-cloud heights and those used in the retrieval. MODIS is also on the Aqua platform and uses a mixture of infrared, near-infrared and visible channels to retrieve cloud optical properties at 10 km resolution. Here we used the CloudTopHeight product from the Collection 6 (MYDO6) dataset; see https://modis-atmos.gsfc.nasa.gov/_docs/C6MOD06OPUserGuide.pdf for details. We have masked the water clouds using the MODIS ice cloud phase product.

The left hand panel shows that the ECMWF ice cloud placement north of the island of Papua is at 8 km (light blue) with a number of high cloud tops straddling the top most part of the granule; the center panel shows our algorithm moved the 8 km high clouds to be northeast of the island, plus it placed some very high cold DCC cloud tops to lie almost on a line along 145 E longitude, which is consistent with what was retrieved by MODIS (right hand panel). In addition clouds over the island of Papua were removed by our algorithm, consistent with what MODIS retrieved.

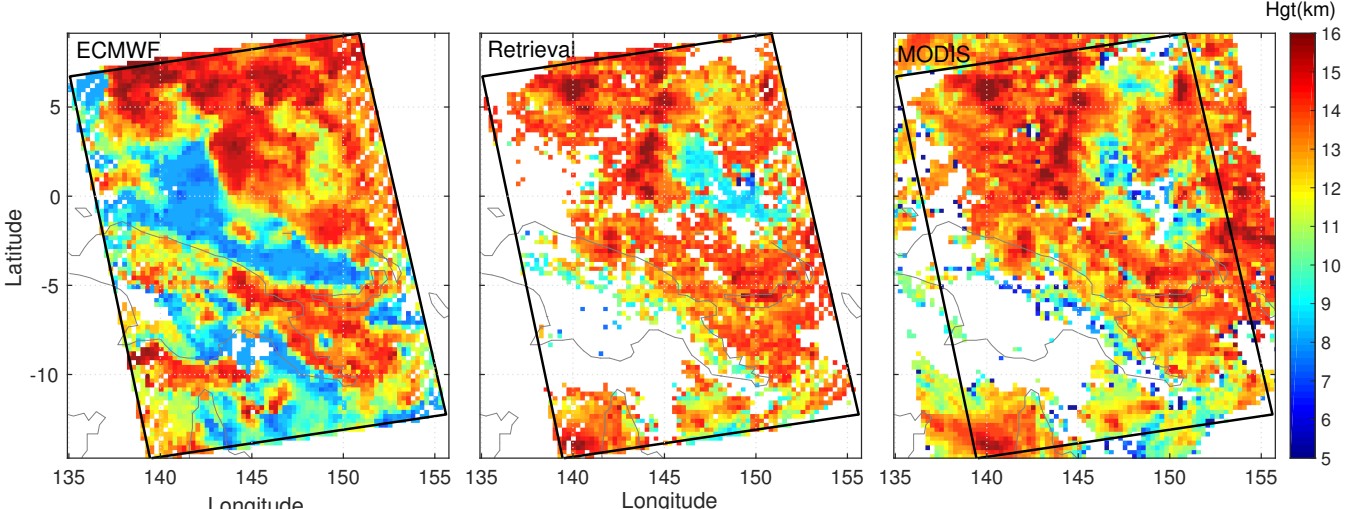

**Figure 17.** Comparisons between cloud top height : (L) Original ECMWF ice cloud top (C) ice cloud top heights used in the retrieval (ice clouds with optical depth < 0.5 have been removed) (R) MODIS L2 ice cloud top heights; the black lines outline AIRS granule 039. It is clearly evident that the initial cloud tops from ECMWF were reset by our algorithm to closely resemble the MODIS retrieved cloud tops; also note the similarity to the observed BT1231 in Figure 9. Generally the low ECMWF cloud heights are associated with very low optical depths.

**7   Conclusions**

A fast infrared radiative transfer algorithm with the ability to handle two scattering layers (from clouds, aerosols, volcanic dust) has been described and compared to a more sophisticated, and often slower approach (Maximum Random Overlap). Our ultimate goal is to perform single-footprint retrievals with hyperspectral IR sounder radiances. In particular we wish to handle the very common case of two cloud layers (water, ice) in order to provide accurate, higher spatial resolution retrievals

of temperature and water vapor (and other minor gases). This approach uses the observed radiances in the retrieval, rather than





derived equivalent cloud-cleared radiances that are presently used for the NASA AIRS Level 2 products. The complexity of true cloud structures cannot be retrieved with hyperspectral IR radiances, and we have shown that only a maximum of 2-4 cloud parameters can be derived from a single scene, suggesting that only a simple RTA is needed.

However, if the *a-priori* cloud parameters are not sufficiently accurate, it can be very difficult for the retrieval to converge
quickly, or at all. Our approach uses NWP model fields (here ECMWF) to initialize the cloud model fields. Four sub columns (at most) are needed to compute a radiance for one scene, which is a small speed penalty in fast RT models, where most of the time is spent in computing the atmospheric optical depths. The TwoSlab model can be an order of magnitude faster than typical implementations of MRO and has nearly the same accuracy, both in terms of mean spectral radiances and radiance PDFs. The spectral bias between all-sky AIRS observations and calculations are dominated by spatial location mis-matches between
actual and forecast clouds. Both approaches used the ECMWF cloud fields, and in general both differed from observations similarly. For example, PCRTM/MRO is slightly more accurate in the tropics than SARTA TwoSlab, while the opposite is true in polar regions. However, the comparisons of RTA simulations to observations are both limited by the accuracy of the NWP model fields and especially by small spatial mis-matches between NWP and observed clouds. The larger errors of both RTA approaches in the polar regions indicate that ECMWF clouds have too low optical depths.

We demonstrated the feasibility of the SARTA TwoSlab approach by performing single-footprint retrievals using an AIRS tropical granule. Our approach to the retrieval cloud initialization and *a-priori*, which is key to the successful results shown here, was to use NWP cloud fields in the region of the footprint of interest based on matching simulated and observed radiances. These matched cloud fields are then converted from N-layer NWP cloud fields to the two-layer SARTA Two-Slab. Together these steps provide a method for robust, and fast, retrievals from single-footprint all-sky hyperspectral spectra.

A major advantage of single-footprint retrieval using the OEM approach are the retrieval quality diagnostics that are provided within the OEM framework. We demonstrated that the retrieval DOFs are reduced in the presence of thicker clouds. However, we were able to reproduce much of the water vapor variability in ECMWF (assumed to be relatively accurate, partly because it agrees with our retrievals) when using a climatology for the water vapor and temperature *a-priori*.

Existing AIRS L2 retrievals fail in scenes with thick clouds and where the 3-by-3 set of radiances used for cloud-clearing
are too homogeneous (which is not always in the case of thick clouds as seen in Figure 12). This is particularly troublesome for long-term climate studies in that the AIRS L2 sampling is incomplete and may alias certain climate variables, especially for water vapor where microwave retrievals (that are part of the AIRS L2 system) have more limited value.

The retrieval approach examined here may be able to address sampling limitations of the existing AIRS retrievals by (1) using single-footprint retrievals that are not affected by cloud-clearing failures for highly homogeneous scenes, and (2) using
*a-priori* information in a statistically correct way under condition of thick clouds. A possible approach is to use a reanalysis for the *a-priori* rather than climatology in order to insert the best possible information in cases where the DOFs of the retrieval are very low.

This work does not represent a rigorous analysis of the accuracy of our retrieval approach, but only a proof-of-principle that the technique appears viable. In particular, the temperature retrievals are not stressed in a tropical environment, although our
results suggest significant skill for water vapor. The retrieval tests shown here were mostly all over ocean where the surface





emissivity is well known. Over land, we will need to include a variable surface emissivity into the retrieval. The time taken to retrieve one single footprint (at the 100 layer native SARTA vertical resolution) is on average under 2.5 seconds, which included matching the AIRS L1b radiances to climatology and NWP cloud fields and converting the NWP cloud profiles to slab clouds. This is very competitive with the official AIRS L2 product which takes about 1.5 second per Field of Regard using

trapezoid vertical function (but does retrieve profiles of some additional trace gases and computes Outgoing Longwave Radiation).

## 8    Acknowledgements

We acknowledge the use of ECMWF model fields to compute radiances. The hardware used in the computational studies is part of the UMBC High Performance Computing Facility (HPCF). The facility is supported by the U.S. National Science

Foundation through the MRI program (grant nos. CNS–0821258 and CNS–1228778) and the SCREMS program (grant no. DMS-0821311), with addtional support from the University of Maryland, Baltimore County (UMBC). See www.umbc.edu/hpcf for more information on HPCF and the projects using its resources. Helpful discussions with George Aumann and Eric Maddy are gratefully acknowledged. Scott Hannon wrote the clear and scattering versions of SARTA and helped develop the TwoSlab clouds.

## 9    Appendix I: Clear-Sky Retrieval Comparisons with ECMWF

Clear-sky biases are likely to arise from inaccuracies in the geophysical parameters, such as highly variable water vapor fields and surface temperatures. The radiance measured by/simulated for the 1231 cm$^{-1}$ channel for clear-sky scenes over the oceans is dominated by the surface temperature, water vapor (which is very variable) and to a much lesser extent temperatures in the lower atmosphere; errors in any of these would affect the comparisons of observed versus simulated radiances. Using an

OEM retrieval scheme (Rodgers, 2000) (also see Section 6), we investigated possible errors for NWP fields used in clear-sky scene calculations by using the AIRS thermal infrared window channels to retrieve tropical sea surface temperature (SST) and column water vapor (WV) amounts, as well as column $O_3$ amount using the 10 $\mu$m channels. Averaged over ~10000+ Fields of View (FOVs) for day and for night, the nighttime ECMWF SST was adjusted by an offset of -0.5 ±2.6 K, while the column WV was adjusted by a multiplicative factor of 1.1 ±0.2 and the column $O_3$ was adjusted by a multiplicative factor of 1.2 ±0.05.

The corresponding daytime adjustments were -0.3 ±0.8 K, and 1.1 ±0.1 for column WV and 1.12 ±0.05 for column $O_3$. While a discussion of the SST adjustments is outside the scope of the paper, the required reduction from the retrieval suggests there is some residual cloud leakage present in the AIRXBCAL clear-sky dataset.

## 10    Appendix II: Information Content of AIRS Radiances

An uniform mixing ratio ice cloud (10e-6 g/g) from 200 mb to 440 mb and an uniform mixing ratio water cloud (1e-6 g/g) from

440 mb to 900 mb were inserted into a tropical profile spanning 97 AIRS layers (1013 mb to TOA). Finite difference jacobians





for the temperature, humidity and cloud profiles were used to compute the degrees of freedom of signal (Rodgers, 2000); AIRS noise equivalent change in radiance (NeDN) converted to BT noise levels were used for a diagonal noise $S_e$ matrix. For this Appendix we use a diagonal $S_a$ geophysical error matrix which had a 1.0 K temperature error and 0.1 fraction (10%) WV(z) error at all layers; similarly we assumed a 0.1 fraction (10%) error for CIWC(z), CLWC(z) at all layers. The computed degrees

of freedom of signal for T(z), WV(z), CLWC(z), CIWC(z) were [13.78, 6.46, 1.75, 2.45] respectively. The last two numbers imply that the information in the 100 layer cloud profiles can indeed be compactly represented by two parameters (cloud top, cloud amount). The corresponding numbers computed for a clear atmosphere are [13.72, 6.99, 0, 0], while those obtained for a thick ice cloud (DCC) and thick water cloud atmosphere are [7.89, 0.99, 0, 2.41] and [10.34, 3.52, 2.30, 0] respectively.

## 11    Appendix III : Sensitivity Analysis of the TwoSlab cloud scheme

The SARTA TwoSlab model has four parameters per cloud, plus a cloud slab fraction and cloud slab overlap parameter that are derived from NWP model fields. The four parameters are the vertical placement and width of the slabs, the cloud loading (integrated CIWC or CLWC amounts), and the effective particle size. Since there are only 2-4 degrees of freedom for clouds in the spectra, for the retrieval only the cloud amounts were varied while the vertical placement, fraction and effective particle size were kept fixed, after the "closest" cloud was found.

Here we briefly explain the changes in the simulated radiances as the cloud vertical placements are changed. An observation dataset of 7377 AIRS observations from 2009/03/01 is used here, as it was chosen to provide maximum variability due to clouds, over land and ocean, and span all climate regions (personal communication, George Aumann, Jet Propulsion Laboratory, CA). The BT1231 cm$^{-1}$ channel is used to study differences between observations and calculations. For the TwoSlab model two placings of the cloud slabs were studied - one where the slab was placed where the cloud's weighting function peaks

(P), and the other at the NWP cloud profile centroid (C).

As can be seen from the whisker plots of Figures 6 (left panel) and 7 (all three sub-panels), especially when considering the mean and standard deviation, the MRO calculations are more similar to the TwoSlab (C)entroid calculations than the TwoSlab (P)eak calculations. This can be understood from the point of view of where the cloud radiates from : in the (P)eak case we place the cloud higher up, which leads to colder calculations; in the (C)entroid case one would expect the multiple sub-pixels of a MRO simulation also to radiate primarily from this region. A (finite difference or analytic) jacobian is easily computed using

the slab clouds, while a jacobian with the MRO representation would be computationally expensive, and probably ill-defined as the sub-pixel cloud amounts and fractions are randomly determined at each stage of the calculation. On average, placing the clouds at the (C)entroid globally displaces water clouds downwards by about 80 mb (from 723 mb) and ice clouds downwards by about 60 mb (from about 400 mb) from the (P)eak cases.



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
