# Peer review of "Single Footprint Retrievals for AIRS using a Fast TwoSlab Cloud-Representation Model and All-Sky Infrared Radiative Transfer Algorithm"

_Atmospheric Measurement Techniques, 2017_

## Referee Comment (RC1) · Anonymous Referee #1 · 26 Sep 2017

General comments:

Most operational hyperspectral retrieval systems perform cloud-clearing of the observed radiances before the atmospheric parameters are retrieved. The process of cloud-clearing has some significant drawbacks; it not only introduces errors (and fails under certain cloud conditions) but also reduces the spatial resolution from single footprint resolution to a 3x3 footprint array. Larger footprints prevent accurate retrieval of small-scale atmospheric features and parameters, which are highly variable such as water vapor. Deriving accurate temperature and sounding profiles under cloudy condi-

tions is still a challenging task, but detailed information about the vertical atmospheric structure for every single footprint is absolutely necessary to have a positive impact in weather, environmental and climate applications. This paper describes a retrieval method, to be used eventually in operations, that uses a forward model with a two-slab cloud presentation and an optimal estimation technique to retrieve temperature and humidity profiles for every single footprint. A detailed description of the retrieval approach and its proof-of-concept is given, and promising results are shown, evaluated and discussed adequately. I recommend the paper for publication after minor revision, which should include a careful rewrite of the material in more concise, coherent sentences and paragraphs.

Specific Comments:

Page 2, lines 4-8: In addition to (or instead of) citing [Weisz et al. 2007] I suggest the following more recent paper

Weisz, E., W. L. Smith Sr., and N. Smith (2013), Advances in simultaneous atmospheric profile and cloud parameter regression based retrieval from high-spectral resolution radiance measurements, J. Geophys. Res. Atmos., 118.

Furthermore, Kahn et al. (2014), who also performs cloud parameter retrievals on individual scenes, should also be mentioned here first rather than on page 30.

Regarding climate studies, a publication worth mentioning is

Smith, N., W. L. Smith, E. Weisz, and H. E. Revercomb (2015), AIRS, IASI, and CrIS retrieval records at climate scales: An investigation into the propagation of systematic uncertainty, J. Appl. Meteorol. Climatol., 54, 1465–1481.

which describes change in the climate system using single field-of-view hyperspectral retrievals under all sky conditions.

Section 2.1: Please state which AIRS channel property file you are using to extract the 'good' channels as well the corresponding NEDT values (shown in Fig. 2)

Section 2.3.2: Can you state what version of PCRTM is used here?

Section 4.2: the motivation for using the 1231 cmˆ-1 channel in the figures and results that follow should be clearly stated here first. It would be also useful to state the corresponding wavelength and the MODIS band equivalent.

Technical Comments:

Please make sure the manuscript undergoes thorough and careful editing. Many sentences are too long, unclear and/or confusing, and the usage of abbreviations, parentheses, spacing, etc. is inconsistent. For instance,

Page 3, line 3: remove 'could'

Page 3, lines 13-15: this sentence is unclear, please rewrite

Page 6, line 13: use 'added' instead of 'adding'

Page 6, line 26: add space after 'does'

Page 7, line 3: add space after the comma in (CLWC, CIWC)

Page 11, line 18: use lower-case "L" in 'TwoSLab'

Page 11, lines 19-21: should be 'become' (not 'becomes'), 'show' (not 'shows'), and 'are much smaller' instead of 'is much smaller'.

Page 11, line 26: use 'were proportional' (not 'was proportional')

Pages 11, 14, 15 etc.: 'SARTA TwoSlab' or 'SARTA/TwoSlab'? Please use consistent terminology.

Page 13, line 13: remove the comma after 'differences'

Page 13, line 15: inconsistent use of parentheses for in-text citations (throughout the paper)

Page 14, line 10: (PDFs) instead of (PDF)s.

Page 14, line 13: please rewrite 'as is evident in from Figures 4 and 5'

Page 16, line 1: use 'shows' instead of 'plots'

Page 16, line 3: there is a space missing after the comma in (1),(2)

Page 16, line 6-7: suggest using 'decreases' instead of 'lowers'

Page 16, line 21: 'led' instead of 'lead'

Page 17, line 2: use either 'pdfs' or 'PDFs'

Page 17, line 17,18: please rewrite "They could either at too low an . . ."

Page 20, line 23: use 'is a block diagonal matrix' instead of 'is block diagonal'

Page 21, line 3: 'physically-based' (not 'physically-gased')

Page 21, line 29: add full name for MERRA

Page 24, line 9: delete repeated 'after the retrieval'

Page 24, lines 15-16: remove parentheses

Page 27, line 4: remove "]" after 14

---

## Referee Comment (RC2) · Anonymous Referee #2 · 29 Sep 2017

The manuscript Âń Single Footprint Retrievals for AIRS using a Fast TwoSlab Cloud-Representation Model and All Sky Radiative Transfer Algorithm Âż of DeSouza-Machado et al., submitted to Atmopsheric Measurement Techniques presents a study in two parts. The first one compares two infrared cloudy radiative transfer models (RTM), a fast and complex cloudy RTM and a fast and simplified cloudy RTM. Based on their results they used in the second part the fast and simplified cloudy RTM model to improve water vapor and temperature profile retrievals from AIRS cloudy observations. The results presented here are very encouraging and comparison with other

independent product such as NUCAPS will be interesting in the future. Overall, the paper is well written and well structured. I do recommend this manuscript for publication in AMT but after minor revisions.

General comments:

1) It should be mentioned in the title that the All-sky radiative transfer algorithm cover the infrared spectral range.

2) The last sentence of the second paragraph of the introduction seems a bit too simplistic considering the large amount of works done by national weather services to assimilate infrared cloudy radiances in NWP. If the cloud-clearing method is operationally used by NOAA and NASA, other methods such as CO2-slicing, Maximum residual method and 1DVAR are used to characterize single layer cloud for operational application. I suggest the author to provide at least some references to these works.

3) I do not see the utility of Figures 13 and 14. Does the authors want to explain why ECMWF is better than climatology for the retrieval ? If yes, then it must be stated in the text.

Specific comments:

Page 4, line 31: What is the spectral resolution of AIRS? Is the typical 0.2 K noise for cold or hot scenes?

Page 5, line 10: replace 00.00 by 00:00

Page 5, line 12: (latitude,longitudes) can be remove if the unit of 0.25+/-0.05 is given.

Page 6, line 8: The sentence is not precise enough. Do you mean gamma size distribution? If yes then the unit of the effective variance should be added as well as the effective radius or diameter.

Page 6, line 16: Does SARTA use the same refractive indices as PCRTM ?

Page 6, line 26: A space is missing before the reference.

Page 7, line 11: Why cloud content profile are smoothed?

Page 7, line 23: Are case 2 often happen?

Page 7, line 30: What is the justification of adding a random offset to the effective diameter?

Page 9, line 6: I think the word types should replaced by layers if case 2 happen.

Page 11, line 17: What are the standard deviation or RMS of the difference Observation minus simulation? Are they comparable between SARTA and PCRTM?

Page 13, line 15: The first bracket of the second reference is not at the right place.

Page 14, line 3: replace 1 1/2 by 1.5 for consistency

Page 15, line 1: I do not see difference in the slab position between blue and cyan. Can you explain it better?

Page 15, line 5: The first bracket of the reference is not at the right place. I also suggest to refer the listing (1), (2), (3) and (4) to the figure.

Page 16, line 3: Positions (1), (2) and (3) are not represented on the right panel of Figure 6.

Page 17, line 1: Are pdfs normalized? If yes it should be mentioned both in the text and in the figure caption.

Page 17, line 19: Is these interpretations have been already shown by other studies?

Page 17, line 22: I suppose ice contamination is sea-ice?

Page 17, line 28: This sentence seems to repeat the sentence before, please reformulate.

Page 19, line 5: There is again a bracket problem with this reference.

Page 20, line 11: Replace Tikonov by Tikhonov.

Page 20, line 13: Put the references before the dot.

Page 20, line 16: The forward model error has been set to be < 0.2K. This is very optimistic for infrared cloudy simulations. As comparison in figure 5, you found a standard deviation of 20 K when comparing observation with RTM. How do you justify that?

Page 20, line 21: What do you mean with logarithmic multiplier for ozone? What is the unit of the cloud amount?

Page 20, line 25: Please, correct Tikhonov.

Page 20, line 29: How the 10% cloud amount uncertainty has been chosen?

Page 21, line 3: Correct physically-based

Page 21, line 5: Appendix II instead of 10

Page 21, line 9: remove file.

Page 21, line 26: the value of the surface temperature uncertainty is not consistent with line 28 of page 20.

Page 24, line 13: Please explain what is Âńfinal UMBCÂż in the legend of Figure 11?

Page 27, line 4: the DOFS and FOVs numbers are not consistent with those of the figures 13 and 14 labels. Please clarify.

Page 27, line 7: I do not see on the figure where is the little difference between the retrieved profile temperature and the a-priori. If it was the case then the red full line would be close to the 0 line ?

Page 29: please indicates the units of relative humidity both in the figure and in the label

Page 30, line 1: please correct "deg"

Page 30, line 20: this paragraph is difficult to understand since there is no figure to help to reader. Is this a general feature of the retrieval or is this a feature of the track B ?

Page 30, line 30: This result is very interesting and I suggest the author to compare this results with other works (for example the work of Heymsfield, A.J., S. Matrosov, and B. Baum, 2003: Ice Water Path–Optical Depth Relationships for Cirrus and Deep Stratiform Ice Cloud Layers. J. Appl. Meteor., 42, 1369–1390, https://doi.org/10.1175/1520-0450(2003)042<1369:IWPDRF>2.0.CO;2 )
* * *

---

## Author Response (AR1)

Sergio De Souza-Machado, Department of Physics, University of Maryland Baltimore County, 1000 Hilltop Circle, Baltimore MD 21250

November 22, 2017

Dr. Andrew Sayer, Associate Editor, Atmospheric Measurement Techniques

Dear Dr. Sayer,

This cover letter is included with the submission of our revised paper (amt-2017-261) entitled

**"Single Footprint Retrievals for AIRS using a Fast TwoSlab Cloud-Representation Model and All-Sky Infrared Radiative Transfer Algorithm"**

together with our responses to the reviewers.

We thank you for the extension we were given to address the concerns of the referees. We are confident we have taken great care to address them all, and look forward to your favorable response.

Please note that I will be away attending the ITOVS meeting in Germany from Monday November 27 till December 5, 2017. If you wish I can designate Dr. Larrabee Strow as contact while I am away. However he is currently unwell and may not be able to respond. Plus we are right at the beginning of the extended Thanksgiving Holidays. So as needed I would appreciate it if you could please give additional response time to make any fixes (the paper .tex files and figures are on my office computer, and I may have trouble with the internet connections).

In writing the manuscript, as before we have adhered as strictly as possible to the AMT manuscript guidelines. This document includes

- Page 1 : this cover letter
- Pages 2-4 : response to Reviewer 1 (submitted yesterday as amt-2017-261-AC1-supplement.pdf)
- Pages 5-9 : response to Reviewer 2 (submitted yesterday as amt-2017-261-AC2-supplement.pdf)
- Pages 10-11 : list of relevant changes from the above two responses
- Pages 12-51 : output from latexdiff between the original and revised manuscripts

Should you need to contact me, my email address is sergio@umbc.edu.

Sincerely,

Sergio De Souza-Machado

**Single Footprint Retrievals for AIRS using a Fast TwoSlab Cloud-Representation Model and All-Sky Radiative Transfer Algorithm"**

by DeSouza-Machado et. al.

We thank the reviewers for their comments, questions and suggestions to improve the paper. Below we detail our responses to their concerns. For ease of review, we type-faced the reviewers questions in blue. When we refer to pages and line numbers in our answers, the context should make it clear whether we are talking about the original manuscript or our current revised manuscript.

Reviewer 1

Specific comments

1) Page 2, lines 4-8: In addition to (or instead of) citing [Weisz et al. 2007] I suggest the following more recent paper

Thank you for making us aware of this paper, we have replaced the 2007 reference with the 2013 reference. Page 2, lines 4-8 now read as follows

"Earlier single footprint retrievals using eigenvalue regression methods have been used with these all-sky (cloud and clear) radiances (see for example Weisz et al. (2013)) ..."

2) Furthermore, Kahn et al. (2014), who also performs cloud parameter retrievals on individual scenes, should also be mentioned here first rather than on page 30.

We have modified the manuscript by moving lines from (old manuscript) Page 30, lines 28-30 to page 3, Lines 2-5 (new manuscript)

"We note here the regression based single footprint retrievals provide cloud top information; similarly cloud phase and cirrus effective diameter and optical thickness retrievals are generated at AIRS single footprint resolution (Kahn:2014) after the L2 thermodynamic retrievals are done, in a separate step that keeps all other retrieval variables constant."

3) Regarding climate studies, a publication worth mentioning is Smith et. al (2015), which describes change in the climate system using single field-of-view hyperspectral retrievals under all sky conditions.

We have added the following sentence towards the end of the Conclusions section (Page 33, Lines 25-27)

"Smith:15 discusses climate change studies using the homogeneous geographic sampling resulting from single footprint retrievals that are physical-statistical based (whereas ours are physical based, using an allsky RTA through all the iterations)."

**4) Section 2.1: Please state which AIRS channel property file you are using to extract the good channels as well the corresponding NEDT values (shown in Fig. 2)**

AIRS channels that have remained stable over the life of the mission were selected for these retrievals to allow accurate trend studies in future work. These channels were further filtered by us after looking at the time series of over-ocean biases between AIRXBCAL (which contains scenes deemed to be clear for each day) scenes and simulated radiances from the ECMWF ERA-Interim reanalysis.

We have re-arranged Section 2.1 and added the following paragraph

"About 1500 AIRS channels that have remained stable over the life of the AIRS mission were selected for this paper. retrievals. This was done by examining the statistics of the 14+ time series of AIRS radiances (of all channels) in the AIRXBCAL clear-sky data set (ocean scenes only), which contains scenes deemed to be clear for each day. More details about this channel list can be obtained from the authors. NeDT values used in this paper come from the v9.5.0 (2011/07/01) file available at https://disc.gsfc.nasa.gov/information/documents?title=AIRS

**5) Can you state what version of PCRTM is used here?**

We used v2.1, that information has been added to Page 6, Line 27. A later version has Nonlocal thermodynamic equilibrium added on for the 4 um channels, and improved solar scattering computations. However we avoided using those short wave channels throughout the paper, and so this should not affect the results presented here.

6) Section 4.2: the motivation for using the 1231 cm-1 channel in the figures and results that follow should be clearly stated here first. It would be also useful to state the corresponding wavelength and the MODIS band equivalent.

Old manuscript, Section 5, Page 13, Lines 1-3 have now been moved towards the end of Section 4.2 (Page 12, Lines 3-6). We have added in the wavelength information, but have chosen not to provide the MODIS band equivalent since that instrument's Channel 5 spans 1230-1250 cm-1 and contains many weak water lines. Conversely AIRS Channel 1291 (centered at 1231 cm-1) spans about 1 cm-1 and so is mainly affected only by the water continuum.

Technical Comments

1) Page 3, line 3: remove could

Fixed

2) Page 3, lines 13-15: this sentence is unclear, please rewrite Fixed, these lines have been changed to Page 3, Lines 15-20

"The OEM methodology provides the user with objective diagnostic information, such as error estimates of the retrieved profiles, Averaging Kernels (AKs) and the information content of the measurements *via* number of Degrees of Freedom (DOF). For example we show later in this paper that our single footprint retrievals have much lower DOFs under thick clouds than in almost clear scenes, which means our retrieval mostly returns the *a-priori* below thick clouds, and can only adjust the profile above such clouds."

3) Page 6, line 13: use added instead of adding

We have rewritten this sentence as "PCRTM calculates reflectance and transmittance of water and ice clouds using a parametrization scheme (Liu et al., 2009) based on a look-up-table trained using 32-stream Discrete Ordinates Radiative Transfer Program for a Multi-Layered Plane-Parallel Medium (DISORT) (Stamnes et al., 1988) and using single scattering properties calculated by Yang et al. (2002), Wei et al. (2004), Huang et al. (2004), and Niu et al. (2007)"

4) Page 6, line 26: add space after does Fixed

5) Page 7, line 3: add space after the comma in (CLWC, CIWC) Fixed

6) Page 11, line 18: use lower-case L in TwoSLab Fixed

7) Page 11, lines 19-21: should be become (not becomes), show (not shows), and are much smaller instead of is much smaller. Fixed

8) Page 11, line 26: use were proportional (not was proportional) Fixed

9) Pages 11, 14, 15 etc.: SARTA TwoSlab or SARTA/TwoSlab? Please use consistent terminology. Fixed, now consistently SARTA/TwoSlab

10) Page 13, line 13: remove the comma after differences Fixed

11) Page 13, line 15: inconsistent use of parentheses for in-text citations (throughout the paper) Fixed

12) Page 14, line 10: (PDFs) instead of (PDF)s.

Fixed

13) Page 14, line 13: please rewrite as is evident in from Figures 4 and 5

Fixed

14) Page 16, line 1: use shows instead of plots Fixed

15) Page 16, line 3: there is a space missing after the comma in (1),(2) Fixed

16) Page 16, line 6-7: suggest using decreases instead of lowers Fixed

17) Page 16, line 21: led instead of lead Fixed

18) Page 17, line 2: use either pdfs or PDFs Changed to PDFs

19) Page 17, line 17,18: please rewrite They could either at too low an . . .

Changed to "They could either be at too low an altitude, or they could be at the right altitude and either have low optical thickness or a low cloud fraction."

20) Page 20, line 23: use is a block diagonal matrix instead of is block diagonal Fixed

21) Page 21, line 3: physically-based (not physically-gased) Fixed

22) Page 21, line 29: add full name for MERRA

Fixed, and added reference to Page 23, Line 8 : "Modern Era Retrospective-analysis for Research and Applications (MERRA) (Gelaro et al., 2017))"

23) Page 24, line 9: delete repeated after the retrieval Fixed

24) Page 24, lines 15-16: remove parentheses Fixed

25) Page 27, line 4: remove ] after 14 Fixed

**Reviewer 2**

**General comments**

1) It should be mentioned in the title that the All-sky radiative transfer algorithm cover the infrared spectral range.

Fixed

2) The last sentence of the second paragraph of the introduction seems a bit too simplistic considering the large amount of works done by national weather services to assimilate infrared cloudy radiances in NWP. If the cloud-clearing method is operationally used by NOAA and NASA, other methods such as CO2-slicing, Maximum residual method and 1DVAR are used to characterize single layer cloud for operational application. I suggest the author to provide at least some references to these works.

As noted in the document (Page 2, line 7-8) NWP centers currently only assimilate clear sky infrared radiances - they determine cloud top altitudes, and then assimilate the radiances of channels whose weighting functions peak above these cloud tops. Assimilating allsky infrared radiances is still a work in progress. We have rewritten that sentence and included the following two references at the end (Page 2, line 8) : "In addition for any given scene, from a pre-determined subset of IR sounder channels, Numerical Weather Prediction (NWP) centers generally only assimilate the radiances that have been deemed unaffected by clouds."

Reale, O., K.M. Lau, J. Susskind and R. Rosenberg (2012), AIRS impact on analysis of an extreme rainfall event (Indus River, Valley, Pakistan 2010) with a global data assimilation and forecast system, J. Geophys. Res., 117, DOI: 10.1029/2011JD017093.

Bauer, P., T. Auligne, W. Bell, A. Geer, V. Guidard, S. Heilliete, M. Kazumori, M.-J. Kim, E. H.-C. Liu, A.P. McNally, B. MacPherson, K. Okamato, R. Renshaw, L.-P. Riishojgaard (2011), Satellite cloud and precipitation assimilation at operational NWP centres, DOI:10.1002/qj.905

3) I do not see the utility of Figures 13 and 14. Does the authors want to explain why ECMWF is better than climatology for the retrieval ? If yes, then it must be stated in the text.

This has already been partially covered in the first paragraph of section 6.1.2 : you need to start with a first guess that is as close to the actual state as is possible. Our work with comparing clear sky subsets of AIRS radiances, against radiative transfer calculations using ECMWF model thermodynamic fields, as well as other comparisons against radiosondes, demonstrates that even with space/time mis-matches these ECMWF thermodynamic fields are quite accurate. Conversely for the granule used in the paper as our retrieval demonstration, there were many local convective regions which would not be in the climatology. Figure 15 shows that climatology was very smooth and did not have any of the structure seen by the AIRS L2 retrievals or our retrievals, or that was in the ECMWF model fields.

We have added/changed the following sentences in Section 6.1.2 (bottom/top of Pages 22/23) "We point out that the thermodynamic fields from ECMWF 3-hour forecasts (and/or analysis) are nearly identical to global radiosonde measurements (see for example the figures in Section 3 of [Ingleby, 2017]), and would also be an ideal starting point for the temperature and humidity profiles. However for this "proof-of-concept" paper the temperature and water vapor profile linearization point and *a-priori* is instead taken from a climatology in order to more easily demonstrate the performance of the retrieval algorithm and the cloud and thermodynamic information contained in the AIRS radiances. "

**Specific comments**

1)Page 4, line 31: What is the spectral resolution of AIRS? Is the typical 0.2 K noise for cold or hot scenes?

The nominal resolution is  $\nu/\delta\nu \sim 1200$  so for example FWHM  $\sim 0.5, 1.0, 2.0 \text{ cm}^{-1}$  at 600,1200,2400 cm-1 respectively. We chose to add that the NeDT is 0.2 K noise at 250K on Page 5, line 5

2) Page 5, line 10: replace 00.00 by 00:00 Fixed

3) Page 5, line 12: (latitude, longitudes) can be remove if the unit of 0.25 + / -0.05 is given. Fixed

4) Page 6, line 8: The sentence is not precise enough. Do you mean gamma size distribution? If yes then the unit of the effective variance should be added as well as the effective radius or diameter. Agreed, this line is fixed to now read "water cloud scattering parameters are computed using Mie scattering coefficients using water refractive indices from the Optical Properties of Aerosols and Clouds (OPAC) database. The parameters are integrated over a modified gamma droplet size distribution of effective variance 0.1 (dimensionless), and effective radius (typically) of 20  $\mu m$ "

5) Page 6, line 16: Does SARTA use the same refractive indices as PCRTM ? See above

6) Page 6, line 26: A space is missing before the reference. Fixed

**7) Page 7, line 11: Why cloud content profile are smoothed?**

The profiles can have a lot of vertical structure and are sometimes multi peaked; we wanted to smooth out fine structure and so have a cleaner profile for our algorithm to locate the slabs. As is mentioned in the manuscript, we do have a lot of flexibility in the final slab placement. The relevant sentence (Page 7, Line 26) has been changed to "is first smoothed in order to make it easier to localize the positioning of the (ice or water) cloud slabs."

8) Page 7, line 23: Are case 2 often happen?

A random check of our data shows 5-10% of the cloud profiles were reduced so both slabs were water (typically over tropical regions) and 1-4% of FOVS had both clouds as ice (typically over the polar regions).

9) Page 7, line 30: What is the justification of adding a random offset to the effective diameter? Water cloud effective diameters vary with season and geographic location. Without getting too much into details we wanted a first cut at modeling this, and plan to be more systematic in the future. We have changed Page 9, Line 9-10 to reflect this and added a reference

"Water cloud droplet effective diameters vary with season and geographic location (King et al., 2013); to model this we use an effective diameter of 20 um plus a uniformly distributed random offset."

10)Page 9, line 6: I think the word types should replaced by layers if case 2 happen.

We agree this could be better written, and changed "types" to "slabs". We also noticed we had an indexing problem in the same line and now use  $cx_j$ , j = 1, 2 (and now explicitly state *i* is the channel index)

11)Page 11, line 17: What are the standard deviation or RMS of the difference Observation minus simulation? Are they comparable between SARTA and PCRTM?

Table 1 gives very representative numbers for the case of all night time observations (standard deviations of about 11 K). For this subset case of 1000 scenes (which as explained in Appendix III of the text were explicitly chosen for cloud variability), the standard deviations between obs and (SARTA/TwoSlab or PCRTM/MRO) are also very similar (about 22 K); we have now included this information.

12) Page 13, line 15: The first bracket of the second reference is not at the right place. Fixed

13) Page 14, line 3: replace 1 1/2 by 1.5 for consistency Fixed

14) Page 15, line 1: I do not see difference in the slab position between blue and cyan. Can you explain it better?

We assume the referee is looking at Figure 6. There indeed are differences between the blue (P = peak) and cyan (C = centroid) curves. We have slightly changed the wording of the final paragraph of Page 16, adding Lines 8-13 "This panel magnifies the differences shown in the left hand panel. For example when the observed clouds are cold (high clouds), one would expect placing the (ice) slab cloud would produce as high as possible (P) would produce a smaller bias than if you placed the slab cloud lower down in the atmosphere, at the centroid (C). Indeed this is clearly seen in the right hand panel - the blue (peak) bias for the cold clouds (BT  $1231 \le 250$  K) is noticeably less than the cyan (C) bias."

15) Page 15, line 5: The first bracket of the reference is not at the right place. I also suggest to refer the listing (1), (2), (3) and (4) to the figure. Fixed

16) Page 16, line 3: Positions (1), (2) and (3) are not represented on the right panel of Figure 6. Fixed

17) Page 17, line 1: Are pdfs normalized? If yes it should be mentioned both in the text and in the figure caption.

Correct, thanks for pointing out this omission

18) Page 17, line 19: Is these interpretations have been already shown by other studies? Not as far as we are aware of. More groups are now producing scattering infrared RTAs, so this could be one of the topics of future allsky RTA inter-comparisons studies.

19) Page 17, line 22: I suppose ice contamination is sea-ice? Correct, thanks for pointing this out, we added the clarification

20) Page 17, line 28: This sentence seems to repeat the sentence before, please reformulate. We have rewritten this (Page 19, Line 14-16) as "The calculations for the polar regions are noticeably warmer than the polar observations, with the SARTA/TwoSlab and PCRTM/MRO clouds simulations much more similar to each other than to the observations."

21) Page 19, line 5: There is again a bracket problem with this reference. Fixed

22) Page 20, line 11: Replace Tikonov by Tikhonov.

Fixed

23) Page 20, line 13: Put the references before the dot. Fixed

24) Page 20, line 16: The forward model error has been set to be  $\leq 0.2$ K. This is very optimistic for infrared cloudy simulations. As comparison in figure 5, you found a standard deviation of 20 K when comparing observation with RTM. How do you justify that?

The 20 K standard deviation mentioned in Figure 5 arises is a consequence of the above mentioned cloud mismatch between observations and NWP model fields, so is not a forward model error.

As noted in Comment 18 above, there are now a number of RTAs capable of handling cloudy calculations, but they have different cloud representations (for example Maximum Random Overlap, Exponential Random Overlap, our TwoSlab approach and so on). Plus they use different cirrus/water scattering parameters. Due to the time mis-match between observations and NWP fields, no definite study has/can be made, about which is the most appropriate cloud representation that reconciles hyperspectral infrared observations with cloudy RTA calculations. A future intercomparison of these codes against in-situ cloud observations would be beneficial.

In any case, we run into the problem of the practicality of these complex cloud representations for use in a physical retrieval algorithm. Our cloud model is simple, fast and accurate and can easily be used to produce jacobians. The PCRTM/MRO code has been extensively validated against LBLRTM/DISORT; this paper shows good agreement between PCRTM/MRO and SARTA/TwoSlab. Figure 3 shows that when both SARTA and PCRTM use TwoSlab clouds (ie same cloud representation but different scattering algorithms), the biases are on the order of 0.5 K and the standard deviations are on the order of 1.5 K.

Also noted in the paper is that in the thermal infrared there are very few degrees of freedom for clouds. Coupled with the variety and complexity of cloud representations, even if the noise level used in this paper are adjusted, they will not adversely change the essence of the results.

25) Page 20, line 21: What do you mean with logarithmic multiplier for ozone? What is the unit of the cloud amount?

If you take the multiplier for the original profile as unity, and you add/subtract perturbations that come out of the OEM formalism, you run into the danger of accumulated subtractions for the profile multiplier becoming less than zero, hence giving unphysical profiles. If instead you use a logarithmic multiplier, then even if the OEM method gives negative deltas, you have to take the exponential before applying the multiplier, which will always be larger than zero.

In this paper we use integrated cloud loadings  $(g/m^2)$ , as noted in Section 3.1

26) Page 20, line 25: Please, correct Tikhonov. Fixed

27) Page 20, line 29: How the 10 % cloud amount uncertainty has been chosen?

We base this on looking at the final retrieved cloud amounts versus the cloud amounts we started with (after the cloud swap/initialization). As noted in the manuscript this is a proof-of-concept; in the future we will probably allow for the adjustment of cloud particle size and cloud top as well.

28) Page 21, line 3: Correct physically-based Fixed

29) Page 21, line 5: Appendix II instead of 10 Fixed

30) Page 21, line 9: remove file. Fixed

31) Page 21, line 26: the value of the surface temperature uncertainty is not consistent with line 28 of page 20.

Thanks for pointing this typo. Fixed.

32) Page 24, line 13: Please explain what is "final UMBC" in the legend of Figure 11? That was a mistake, should be "final retrieval"; now fixed (in Figs 11,13,14)

33) Page 27, line 4: the DOFS and FOVs numbers are not consistent with those of the figures 13 and 14 labels. Please clarify.

We mistakenly bracketed the DOFs spanning regimes for the plots and the text differently; we have now fixed the text to agree with the figures.

34) Page 27, line 7: I do not see on the figure where is the little difference between the retrieved profile temperature and the a-priori. If it was the case then the red full line would be close to the 0 line ?

We have rephrased the sentence (Page 27, lines 5-10) to read "The low DOF case shows a smaller difference between retrieved profile temperature and a-priori in the lower troposphere, compared to the free and upper troposphere."

35) Page 29: please indicates the units of relative humidity both in the figure and in the label

Fixed

36) Page 30, line 1: please correct deg Fixed

37) Page 30, line 20: this paragraph is difficult to understand since there is no figure to help to reader. Is this a general feature of the retrieval or is this a feature of the track B ?

This is a general feature of the retrieval for this granule. Basically NWP models underestimate deep convective cloud tops. So in order to produce calculations which are cold, our cloud initialization needs to match colder observations with cloud fields that have a lot of ice cloud associated with them; increasing the ice cloud fraction means the observed emission is now better localized as coming from these cold cloud tops.

38) Page 30, line 30: This result is very interesting and I suggest the author to compare this results with other works (for example the work of Heymsfield, A.J., S. Matrosov, and B. Baum, 2003: Ice Water PathOptical Depth Relationships for Cirrus and Deep Strati- form Ice Cloud Layers. J. Appl. Meteor., 42, 13691390, https://doi.org/10.1175/1520-0450(2003)

Thanks for pointing out this paper to us. We seem to be within a factor of two of the OD:IWP ratio in the paper you mentioned. This could be explained by differences in the cirrus habit scattering parameters used by them versus those used by us. In addition the amount of cloud (or aerosol) loading required to minimize infrared biases depend sensitively on height. Exploring this further is outside the intent of the paper, but we will certainly keep this in mind for future studies when for example we could include more parameters in the state vector (such as including ice cloud effective particle size and cloud top height in the retrievals) in order to have the best possible estimate of cloud loading. List of all relevant changes

These refer to document made by "latexdiff". Note that it was not correctly cross-referencing the original document, so there are many examples of Figure ?? being fixed to eg Figure 10

- Title (page 1) : added "Infrared"
- Page 2, lines 3-9 : paragraph was changed to include "Earlier single footprint retrievals ...(Weisz et al, 2013)" . In addition end of paragraph reinforces the fact that NWP centers assimilate clear sky radiances, and references added.
- Page 3, lines 2-5 : were moved from later in the document, and describe cloud information retrieved from hypersepctral sounders, as documented in earlier work by Wesiz (2013) and Kahn (2014)
- Page 3, Lines 16-21 include the re-write of the built-in diagnostics that the OEM formalism offers.
- The whole document now uses SARTA/TwoSlab
- Page 5, Section 2.1 : includes the 250 K scene noise, plus a new paragraph (lines 10-15) describing how we obtained the list of good channels and the AIRS noise file.
- Page 5, lines 22,24 : minor changes for the time (00:00 GMT) and the latitude/longitude grids
- Page 6, Lines 22-25 : rephrased the sentence(s) describing how we computed the water cloud scattering coefficients
- Page 6, Lines 30-32 (and Page 7, line 1-2) : rephrased the sentence describe the implementation of DISORT scattering into PCRTM
- Page 7, lines 29-30 : rephrased why we smoothed the cloud profiles prior to finding the cloud slab position
- Page 9, lines 11-12 : added more description and a reference as to how we chose water cloud effective diameters.
- Page 9, lines 16-24 : fixed the indexing for the 4 radiance streams
- Page 12, lines 6-13 : Rewrote the paragraph so it includes why we used the 1231 cm-1 channel (moved from later in the pages), and also added standard deviations in the comparisons against observations
- Page 14 : Minor fixes (spacing between text and reference), and figure cross references
- Page 15 : Minor fixes (PDFs), and figure cross references
- Page 16, lines 2-3 : added new sentence, fixed spacing
- Page 17, lines 5-10 : As requested by referee, we detail the differences between the placing of the cloud slabs in (C)entroid versus (P)eak positions.
- Page 18, lines 21-22 : have more clearly phrased the possible reasons why the calculations over polar regions are noticeably different than observations
- Page 20, lines 2-5, line 10 : Made minor fixes as requested by reviewers
- Page 21, line 17 : Tikhonov spelled correctly
- Page 22 : minor fixes to the document as requested

- Page 22 Lines 24, 31-34 : we have now more fully described why it is preferred to start with ECMWF model fields (continues into page 23, 24)
- Page 24, lines 9-10 : MERRA acronym and reference added
- Pages 24-27,29 : Minor fixes to the cross-references
- Pages 28,29 : right hand figure panels now contains the RH units, plus we have replaced "UMBC" with "Final Retr"
- Page 28, Lines 6-10 : Fixed the description for the low DOF case
- Page 31, Lines 27-32 were moved to Page 3, Lines 2-5
- Page 34, Lines 28-30 : added the discussion of the Smith (2015) climate studies paper using hyperspectral sounder and a physical-statistical retrieval.

**Single Footprint Retrievals for AIRS using a Fast TwoSlab Cloud-Representation Model and All-Sky Infrared Radiative Transfer Algorithm**

Sergio DeSouza-Machado1, L. Larrabee Strow1,2, Andrew Tangborn1, Xianglei Huang3, Xiuhong Chen3, Xu Liu4, Wan Wu5, and Qiguang Yang5

1JCET, University of Maryland Baltimore County, Baltimore, Maryland

[revised manuscript text omitted]

where  $f_{clr}$  is the clear fraction,  $ex_i$ , i = 1, 2,  $ex_j$ , j = 1, 2 is the exclusive cloud type i slab j fraction and  $c_{overlap}$  is the cloud 20 overlap between the two cloud typesslabs; the exclusive cloud fraction being related to the cloud fraction via the relationship  $ex_i = c_i - c_{overlap} ex_j = c_j - c_{overlap}$ . The model currently exclusively uses ice or water clouds when computing the radiances  $r_i^{(1)}(\nu), r_i^{(2)}(\nu)$  associated with the cloud typesslabs;  $r_i^{clr}(\nu)$  is the clear sky contribution while  $r_i^{(12)}(\nu)$  is the radiance contribution from the cloud overlap. Since the atmospheric gas optical depth computation dominates the run time, computing four radiance streams is not a speed penalty, and the overall average run time per profile is about double that for a single clear sky 25 radiance computation.

**3.3 Maximum Random Overlap Conversion**

The MRO cloud processing for the PCRTM model is described in Chen et al. (2013), and will only be briefly summarized here. MRO converts the NWP water and ozone levels profiles to 100 layer profiles. For each layer, the cloud ice water content and cloud liquid water content mixing ratios are converted to a cloud optical depth. The optical depths at each layer are summed.

30 Layers above 440 mb are considered ice clouds, and layers in the lower atmosphere are assigned to water clouds (Rossow

and Schiffer, 1983, 1991). The effective water diameter is set at 20  $\mu$ m while the effective ice diameter is again temperature dependent, based on the parametrization in Ou and Liou (1995). The cloud cover profile cc(z) is used to generate 50 sub-columns using MRO (Chen et al., 2013) for which one radiance is computed per sub-column; the final radiance is an average over these sub-columns.

**5 4 Inter-Comparisons of SARTA and PCRTM**

**4.1 Clear-Sky Comparisons**

An earlier inter-comparison of the SARTA and PCRTM clear sky models is presented in Saunders et al. (2007). In this subsection we assess the more recent spectroscopy embedded in the SARTA and PCRTM codes, using ECMWF thermodynamic profiles and surface parameters to compare clear sky radiances computed from the models.

- 10 We use 1600 randomly chosen night time scenes observed by AIRS on 2009/03/01 for an inter-RTA clear sky simulation comparison. The locations span all climate zones over ocean and land, as well as all AIRS scan angles. Night time scenes are used to avoid non-local thermodynamic equilibrium (De Souza-Machado et al., 2007) and solar surface reflectivity during the daytime in the 4  $\mu$ m shortwave region. Both of these effects are handled differently by SARTA and PCRTM and are not relevant to this paper.
- Figure ?? 2 shows the calculated BT biases between SARTA and PCRTM clear sky models along with AIRS noise levels. The top panel shows the mean differences, while the bottom shows the standard deviations. The mean bias between SARTA and PCRTM clear calculations is within AIRS noise levels at all channels, except in the methane region (1300 cm-1) and some channels in the water vapor 6.7 μm region. This is due to differing methane and water vapor spectroscopy and continuum models in these two RTAs. In addition, PCRTM uses a density weighted layer temperature that may introduce differences.
  20 Overall, differences between SARTA and PCRTM effective BTs are typically within AIRS noise levels.

**4.2 All-Sky Comparisons for TwoSlab Clouds**

25

30

Here we compare all-sky radiances computed using SARTA and PCRTM, but use the same TwoSlab cloud representation in both RTAs. This tests the differences in each RTA's underlying scattering algorithm by keeping the cloud representation the same in both. Thus this directly compares the relative accuracy of the PCLSAM scattering algorithm used in SARTA against the DISORT-based scattering used in PCRTM.

The PCLSAM algorithm approximations in SARTA are more accurate for absorptive clouds that are more likely in the mid-IR. However, the DISORT-based scattering in PCRTM is more accurate if the cloud representation is correct. In general it would be reasonable to expect the differences to increase with optical depth and/or cloud fraction. In addition, in the TIR the single scattering albedo of water clouds is generally larger than that of ice, so we would also expect larger differences for water clouds.